# Benchmarking Abstract and Reasoning Abilities Through A Theoretical Perspective

Qingchuan Ma [* 1]   Yuhang Wu [* 1]   Xiawu Zheng [1 2]   Rongrong Ji [1 2]

## Abstract

In this paper, we aim to establish a simple, effective, and theoretically grounded benchmark for rigorously probing abstract reasoning in Large Language Models (LLMs). To achieve this, we first develop a mathematic framework that defines abstract reasoning as the ability to: (i) extract essential patterns independent of surface representations, and (ii) apply consistent rules to these abstract patterns. Based on this framework, we introduce two novel complementary metrics: $\Gamma$ measures basic reasoning accuracy, while $\Delta$ quantifies a model's reliance on specific symbols rather than underlying patterns - a key indicator of true abstraction versus mere memorization. To implement this measurement, we design a benchmark: systematic symbol remapping in rule-based tasks, which forces models to demonstrate genuine pattern recognition beyond superficial token matching. Extensive LLM evaluations using this benchmark (commercial API models, 7B-70B, multi-agent) reveal:1) critical limitations in non-decimal arithmetic and symbolic reasoning; 2) persistent abstraction gaps despite chain-of-thought prompting; and 3) $\Delta$'s effectiveness in robustly measuring memory dependence by quantifying performance degradation under symbol remapping, particularly highlighting operand-specific memorization. These findings underscore that current LLMs, despite domain-specific strengths, still lack robust abstract reasoning, highlighting key areas for future improvement.

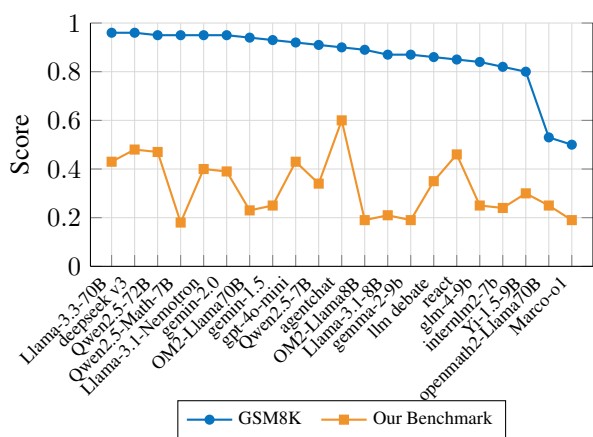

Figure 1. Models often excel on GSM8K (blue line) but show significantly lower performance on our abstract reasoning benchmark (red line), suggesting that domain-specific math tasks may not probe deeper abstract skills.

## 1. Introduction

Abstract reasoning, a cornerstone of human-level intelligence (Holyoak & Morrison, 2012; Penn et al., 2008; Holyoak, 2012; Chollet, 2019; Bober-Irizar & Banerjee, 2024; Xiong et al., 2024), remains a critical yet elusive capability for large language models (LLMs). As philosophically defined, abstract reasoning involves two core processes: abstraction, the extraction of common features from concrete entities (Murphy, 2004), and reasoning, the derivation of new knowledge from existing information (Holyoak & Morrison, 2012). Abstraction provides the fundamental units and organizational structure for cognition, while reasoning operates and infers relationships between these abstractions. Their synergy empowers systems to understand the world and solve problems.

While large language models (LLMs) such as GPT series (Radford et al., 2019; Achiam et al., 2023), Llama seris(Touvron et al., 2023a;b; Dubey et al., 2024), and PaLM-2 (Anil et al., 2023) have exhibited remarkable achievements in various benchmarks, high performance does not necessarily imply *abstract generalization*. Indeed, many benchmark tasks can be tackled via pattern matching

*Equal contribution [1]Key Laboratory of Multimedia Trusted Perception and Efficient Computing, Ministry of Education of China, Xiamen University, 361005, P.R. China. [2]Institute of Artificial Intelligence, Xiamen University.. Correspondence to: Xiawu Zheng <zhengxiawu@xmu.edu.cn>.

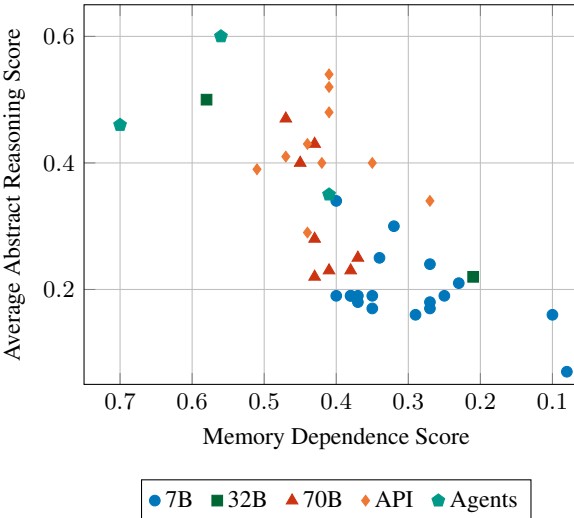

*Figure 2.* Memory Dependence vs. Average Score (Small Scale). The corresponding full-size version is available in Fig.6 of the Appendix.

or memorized heuristics, failing to assess true reasoning.

Numerous benchmarks attempt to assess reasoning, but often fall short for rigorously evaluating abstract reasoning in LLMs. ARC (Chollet, 2019)'s 2D visual format is fundamentally misaligned with LLMs' text-based nature, hindering direct evaluation of their reasoning. Benchmarks like GSM8K (Cobbe et al., 2021) and MATH (Hendrycks et al., 2021), while symbolic, risk measuring memorization of problem-solving heuristics rather than true abstract mathematical understanding. Even BIG-Bench Hard (Suzgun et al., 2023), with its broad task range, lacks a focused theoretical grounding for abstraction, potentially allowing solutions via pattern matching without genuine abstraction. These limitations underscore the urgent need for benchmarks that are both LLM-compatible and theoretically designed to isolate and measure abstract reasoning, moving beyond surface patterns and memorization (Wu et al., 2024; Jiang et al., 2024; Mirzadeh et al., 2024).

Therefore, in this paper, we aim to bridge this gap by constructing a mathematically rigorous theoretical framework to formally define abstract reasoning (Chollet, 2019; Morris et al., 2023) to formally define abstract reasoning as a composite process: first, abstraction as extracting essential patterns from concrete inputs through a mapping function, and second, reasoning as applying consistent rules to these abstract patterns to derive conclusions. Based on this framework, we introduce two complementary metrics to rigorously probe LLMs' abstract reasoning abilities. Specifically, $\Gamma$ measures the basic accuracy of reasoning, reflecting how well models apply rules, and $\Delta$ quantifies a

model's dependence on specific symbols rather than underlying abstract patterns, thus serving as a crucial indicator of genuine abstraction versus mere memorization. Based on these rigorous definitions, we further derive three theorems. This theorem further motivates the core principle for our benchmark design: *systematic symbol remapping in rule-based tasks*, which compels models to demonstrate genuine pattern recognition beyond superficial token matching. Consequently, we design our *domain-general* benchmark, which, unlike domain-specific, math-centric benchmarks like GSM8K, offers a more discerning evaluation through symbolic tasks and abstract rules. This design effectively reduces reliance on memorization and domain knowledge, directly targeting genuine abstract reasoning. We then conduct extensive evaluations of various LLMs (7B-70B, multi-agent) using this benchmark, revealing key limitations and the diagnostic value of $\Delta$.

Our contributions are threefold: (1) a theoretically grounded framework formalizing abstraction and reasoning, and introducing the $\Gamma$ and $\Delta$ metrics; (2) a novel symbolic benchmark with diverse tasks and remapping variants to rigorously test abstract reasoning; and (3) empirical insights from extensive LLM evaluations, revealing persistent challenges in non-decimal arithmetic, symbolic transformations, and function inference, even with Chain-of-Thought and multi-agent methods showing only partial gains. These findings underscore the need for refined benchmarks like ours to guide the development of truly abstractly reasoning AI systems and highlight key areas for future improvement.

## 2. Related Work

**Reasoning Benchmarks.** Existing abstract reasoning benchmarks are often domain-specific, limiting scope and theoretical interpretability (Chollet, 2019; Barrett et al., 2018; Mirzadeh et al., 2024; Frohberg & Binder, 2022; Majumder et al., 2024; Li et al., 2024a; Yuan et al., 2023). **Visual benchmarks** like ARC (Chollet, 2019) and PGM (Barrett et al., 2018) focus on 2D spatial tasks, misaligned with LLMs' text nature. **Mathematical benchmarks** (e.g., GSM8K (Cobbe et al., 2021), MATH (Hendrycks et al., 2021)), while symbolic, may allow memorization of domain patterns without deep abstract reasoning. BIG-Bench Hard (Suzgun et al., 2023) offers broader tasks but lacks a unified theoretical abstraction framework. Our domain-general, theoretically grounded benchmark with symbolic tasks addresses these limitations for rigorous abstract reasoning evaluation in LLMs.

**Multi-Agent Systems and CoT Prompting.** Chain-of-thought (CoT) (Wei et al., 2022) and multi-agent frameworks (ReAct (Yao et al., 2023), AutoGen (Wu et al., 2023), LLM Debate (Du et al., 2023)) aim to enhance LLM reasoning, but evaluations often lack abstract reasoning focus.

While CoT/multi-agent methods improve performance on some reasoning tasks, they don't fundamentally address core abstraction gaps our benchmark reveals, especially for symbolic transformations.

**Theoretical Perspectives.** Broader theoretical underpinnings, particularly in the domain of neural-symbolic learning and reasoning, are extensively reviewed by (Besold et al., 2021). Theoretical frameworks for abstract reasoning (Barrett et al., 2018; Li et al., 2024b; Mitchell et al., 2023; Chollet, 2019; Morris et al., 2023; Huang & Chang, 2023; Boix-Adsera et al., 2023; Jiang et al., 2024; Wu et al., 2024) offer insights but are often conceptual or task-specific. Barrett et al. (Barrett et al., 2018) explore information theory in visual reasoning; Mitchell et al. (Mitchell et al., 2023) emphasize feature selection in abstraction. However, these often lack direct application to LLM abstract reasoning evaluation. Our work builds on these, offering a unified, grounded framework and benchmark for quantitative LLM evaluation, with metrics measuring invariance and generalization. Such quantitative evaluation of genuine abstraction is crucial, especially as the field explores avenues like neuro-symbolic AI to foster more robust reasoning beyond mere pattern matching (d'Avila Garcez & Lamb, 2020). Furthermore, our benchmark's systematic symbol remapping directly probes the generalization of learned patterns and the analogical reasoning capabilities that are increasingly observed in LLMs (Webb et al., 2023).

## 3. Theoretical Foundations on Abstract and Reasoning

In this section, we establish a rigorous theoretical framework for analyzing and evaluating abstract reasoning in LLMs. While prior research (Holyoak & Morrison, 2012; Penn et al., 2008; Murphy, 2004; Holyoak, 2012; Evans, 2010; Holland, 1986) has explored abstract reasoning from cognitive and philosophical perspectives, we advance the field by providing a formal, mathematically grounded framework specifically designed for computational systems. Our framework consists of three complementary components: formal definitions that precisely characterize abstraction and reasoning processes (Section 3.1), theoretical validation that establishes the mathematical soundness of our approach (Section 3.5), and evaluation metrics that enable systematic assessment of abstract reasoning capabilities (Section 3.3).

### 3.1. Formal Definitions: Abstraction and Reasoning

Abstract reasoning relies on the core processes of abstraction and reasoning. Cognitive science defines abstract reasoning as identifying and extracting essential, generalizable patterns from complex instances, effectively filtering irrelevant details to focus on core concepts for understanding and generalization (Murphy, 2004; Holyoak & Morrison, 2012). These "essential patterns" are crucial for understanding, categorization, prediction, and reasoning, contrasting with superficial details that are intentionally discarded (Holyoak, 2012). In our framework, we represent both concrete instances and abstract features as strings, suitable for modeling symbolic information processed by LLMs.

Abstraction is crucial for effective cognition, enabling complexity management and generalization in reasoning. Abstraction, in essence, is information compression and generalization from concrete to abstract string representations, extracting core, reasoning-relevant information. This involves mapping a concrete instance to an abstract feature, aiming to distill it into a concise, generalized form that retains essential patterns for reasoning, while reducing complexity and variability. Building upon established perspectives (Murphy, 2004; Goldstone & Barsalou, 1998; Ross & Spalding, 1994), we formally define abstraction as follows:

**Definition 3.1** (**Abstraction Mapping**). Let $\mathcal{C}$ be the set of concrete instances, each being an individual instance represented as a string of maximum length $l$, such that $\mathcal{C} \subseteq \Sigma^{\leq l}$ ($\Sigma$ is a finite alphabet). Let $\mathcal{A}$ be the set of abstract features, each an abstract feature $a \in \mathcal{A}$ also a string with maximum length $a \leq l$, such that $\mathcal{A} \subseteq \Sigma^{\leq a}$.

Abstraction is a mapping function $f : \mathcal{C} \to \mathcal{A}$.

For example, recognizing a "dog" involves abstraction to generalize across breeds, focusing on essential features like "four-legged" and "mammalian" while disregarding details like "fur color". In our benchmark, a binary addition problem like "01000110 + 00011111" (concrete instance $c$) can be abstracted to "binary addition operation" (abstract feature $a$), focusing on the operation rather than specific operands. Here, $l$ is the maximum length of the problem string, and $a$ is the length of "binary addition" ($a < l$).

Reasoning, in cognitive science, is broadly defined as deriving new information from existing knowledge (Holyoak & Morrison, 2012; Evans, 2010). It involves manipulating abstract representations, identifying relationships, and inferring conclusions. This process can be formalized as applying a rule to an abstract feature to produce a conclusion. Building on this, we formalize the reasoning function as:

**Definition 3.2** (**Reasoning Function**). Let $\mathbb{R}$ be the set of rules, where each rule $r \in \mathbb{R}$ is a string describing an operation, procedure, or pattern, with maximum length $\rho$, such that $\mathbb{R} \subseteq \Sigma^{\leq \rho}$. Let $\mathcal{A}$ be the set of abstract features (Definition 3.1). Let $\mathcal{Q}$ be the set of conclusions, each conclusion $q \in \mathcal{Q}$ also a string with maximum length $q$, such that $\mathcal{Q} \subseteq \Sigma^{\leq q}$.

The reasoning process is formalized by a function $\mathcal{R}e : \mathcal{A} \times \mathbb{R} \to \mathcal{Q}$.

For instance, with the abstract concept "dog" and the rule

"dogs bark," reasoning infers "This dog is likely to bark" upon encountering a new dog. In binary addition, with the abstract feature "binary addition operation" (from $\mathcal{A}$) and a binary addition rule $r \in \mathbb{R}$, $\mathcal{R}e$ yields a conclusion $q \in \mathcal{Q}$ like "01100101" from operands "01000110" and "00011111". Here, $q$ is the maximum conclusion string length.

### 3.2. Two Types of Composite Abstract Reasoning: Rule-Given and Rule-Inductive

Cognitive science identifies rule-based reasoning, or deductive reasoning, as applying pre-established rules to derive conclusions (Evans, 2010; Johnson-Laird, 1999; Liu et al., 2024b; Wang et al., 2024; Boix-Adsera et al., 2023). Rule-given abstract reasoning focuses on scenarios with explicit or known rules, reflecting our ability to use existing knowledge for inference in new situations—a cornerstone of cognition. This type of reasoning can be formalized as a composite process involving abstraction followed by rule application. Building on these insights and deductive reasoning research (Evans, 2010; Johnson-Laird, 1999; Laird, 2019), we formally define the Rule-Given Composite Abstract Reasoning Function:

**Definition 3.3** (**Rule-Given Composite Abstract Reasoning Function**). When rule $r \in \mathbb{R}$ is given, the composite abstract reasoning function, $\mathcal{H}_\mathrm{G}$, represents the complete abstract reasoning process. It is formalized as the composition of abstraction mapping $f$ (Definition 3.1) and reasoning function $\mathcal{R}e$ (Definition 3.2):

$$\mathcal{H}_\mathrm{G} = \mathcal{R}e \circ f. \tag{1}$$

For a concrete instance $c \in \mathcal{C}$ and a given rule $r \in \mathbb{R}$, it is:

$$\mathcal{H}_\mathrm{G}(c, r) = \mathcal{R}e(f(c), r). \tag{2}$$

Here, $c \in \mathcal{C}$ is a concrete instance string (max length $l$), $r \in \mathbb{R}$ is a rule string (max length $\rho$), $f : \mathcal{C} \to \mathcal{A}$ is abstraction mapping, and $\mathcal{R}e : \mathcal{A} \times \mathbb{R} \to \mathcal{Q}$ is reasoning function. $\mathcal{A}$ is the set of abstract features (strings, max length $a \le l$), and $\mathcal{Q}$ is conclusions (strings, max length $q$).

Consider recognizing dog breeds: we apply learned "dogness" rules to identify even unfamiliar breeds—rule-given abstract reasoning in action. In our benchmark, binary addition exemplifies this. For $c =$ "01000110 + 00011111" and a given binary addition rule $r$, $\mathcal{H}_\mathrm{G}$ abstracts $f(c)$ to "binary addition operation", then $\mathcal{R}e(a, r)$ applies $r$, yielding $q =$ "01100101". The rule is provided; the task is abstraction and application.

Rule induction, or inductive reasoning, contrasts with rule-given reasoning by learning general rules from specific experiences (Holland, 1986; Sloman, 1996; Qiu et al., 2024; Li et al., 2024b; Mirchandani et al., 2023; Merler et al.,

2024). It enables new knowledge acquisition, adaptation, and generalization beyond direct experience, moving from observation to rule discovery—crucial for intelligent behavior. This form of reasoning involves inferring a rule from examples and then applying it to new instances. Building on cognitive foundations and inductive learning theories (Holland, 1986; Tenenbaum et al., 2011; Anderson, 1991), we formally define the Rule-Inductive Composite Abstract Reasoning Function:

**Definition 3.4** (**Rule-Inductive Composite Abstract Reasoning Function**). When the rule is initially unknown and inferred from examples, the composite abstract reasoning function, $\mathcal{H}_\mathrm{I}$, incorporates rule induction. Given example set $e = \{(ec_j, eq_j)\}_{j=1}^m$ and new instance $c \in \mathcal{C}$, it is:

$$\mathcal{H}_\mathrm{I}(e, c) = \mathcal{R}e(f(c), \hat{r}), \quad \text{where } \hat{r} = \text{InferRule}(e). \tag{3}$$

Here, $e = \{(ec_j, eq_j)\}_{j=1}^m$ is an example set of instance-conclusion pairs ($ec_j \in \mathcal{C}$, $eq_j \in \mathcal{Q}$). $c \in \mathcal{C}$ is a new concrete instance string. $f : \mathcal{C} \to \mathcal{A}$ is abstraction mapping, $\mathcal{R}e : \mathcal{A} \times \mathbb{R} \to \mathcal{Q}$ is reasoning function, $\hat{r} \in \mathbb{R}$ is the inferred rule string (max length $\rho$), and $\text{InferRule}(e) \to \hat{r}$ is the rule inference mechanism.

Consider learning a new plant species: we induce a rule string from examples of leaf shape, flower color, etc. In our benchmark, inferring operation "#op" from examples like $e = \{$ ("2 #op 3", "5"), ... , ("7 #op 2", "9") $\}$. $\mathcal{H}_\mathrm{I}$ first infers rule $\hat{r} = \text{InferRule}(e)$ (e.g., "addition"), then for $c =$ "5 #op 4", applies $\mathcal{R}e(f(c), \hat{r})$, yielding $q =$ "9". The system induces the rule from examples, then applies it to a new problem.

### 3.3. Measuring Abstract Reasoning: A Two-Metric Approach

Traditional accuracy metrics, while important for evaluating model performance, are insufficient for assessing *genuine* abstract reasoning in LLMs (Wu et al., 2024; Jiang et al., 2024; Mirzadeh et al., 2024; Dentella et al., 2023). As discussed in Appendix A.1, high accuracy can stem from memorizing input-output patterns or exploiting superficial correlations—a *Rule-Given* reasoning form based on pre-learned associations. This memorization-based "reasoning" is brittle, failing to generalize when symbolic representations change, even with constant abstract task structure. To evaluate true *Rule-Inductive* abstract reasoning—rule extraction and application independent of specific symbols—metrics must discern memorization from genuine abstraction. To overcome this limitation and rigorously assess genuine abstract reasoning in LLMs, we propose two complementary metrics: the Abstract Reasoning Score ($\Gamma$) and the Memory Dependence Score ($\Delta$). The Abstract Reasoning Score ($\Gamma$) is designed to measure the basic accuracy of a model on abstract reasoning tasks using original symbols, establishing a performance baseline.

**Definition 3.5** (**Abstract Reasoning Score,** $\Gamma$). Consider a test set $\mathcal{T} = \{(c_i, r_i, q_i)\}_{i=1}^N$ of $N$ independent tasks. For each task $i$, $c_i \in \mathcal{C}$ is a concrete instance string (max length $l$), $r_i \in \mathcal{R}$ is a rule string (max length $\rho$), and $q_i \in \mathcal{Q}$ is the ground truth conclusion string (max length $q$).

The Abstract Reasoning Score, $\Gamma$, is calculated as:

$$\Gamma = \frac{1}{N} \sum_{i=1}^N \mathbf{1}\big[\hat{H}(c_i, r_i) = q_i\big], \tag{4}$$

where $\mathbf{1}\big[\hat{H}(c_i, r_i) = q_i\big]$ is an indicator function evaluating to 1 if model $\hat{H}$'s predicted conclusion $\hat{H}(c_i, r_i)$ matches the ground truth $q_i$ (string identity), and 0 otherwise; summation is over all $N$ tasks. $\hat{H}$ is the model's composite abstract reasoning function.

A higher $\Gamma$ (0 to 1) indicates greater average accuracy of $\hat{H}$ on $\mathcal{T}$ with original symbols. $\Gamma$ measures baseline performance on abstract reasoning tasks under standard symbolic conditions, showing model performance with familiar representations, but it does not alone differentiate abstraction from memorization.

To assess model dependence on specific symbols and probe *Rule-Inductive* reasoning, we introduce $\Delta$. $\Delta$, the Memory Dependence Score, is designed to quantify a model's dependence on specific symbols rather than underlying abstract patterns. The core idea is to disrupt memorization via symbol mapping $M$, which systematically re-labels symbols in concrete instance strings $c_i \in \mathcal{C}$ of test set $\mathcal{T}$. $M$ preserves abstract task structure while altering surface symbols (e.g., in binary arithmetic, remap '0', '1' to 'A', 'B'), forcing reasoning about binary operations, not memorized '0', '1' associations.

**Definition 3.6** (**Memory Dependence Score,** $\Delta$). Apply symbol mapping $M$ to each $c_i$ in original test set $\mathcal{T} = \{(c_i, r_i, q_i)\}_{i=1}^N$, creating symbol-mapped set $M(\mathcal{T}) = \{(M(c_i), r_i, M(q_i))\}_{i=1}^N$. Rule strings $r_i$ remain in the original symbolic space.

Evaluate model $\hat{H}$ on $M(\mathcal{T})$, calculating accuracy $\Gamma_M$, similar to $\Gamma$:

$$\Gamma_M = \frac{1}{N} \sum_{i=1}^N \mathbf{1}\big[\hat{H}(M(c_i), r_i) = M(q_i)\big]. \tag{5}$$

The Memory Dependence Score, $\Delta$, is the difference between $\Gamma$ and $\Gamma_M$:

$$\Delta = \Gamma - \Gamma_M. \tag{6}$$

$\Delta$ quantifies performance degradation under remapping. Larger $\Delta$ indicates significant performance drop, suggesting high dependence on original symbols and memorization.

Smaller $\Delta$ signifies robustness to remapping, indicating less token dependence and more abstract, rule-based reasoning. Ideally, a truly abstract reasoning model has $\Delta \approx 0$, showing invariance to symbol remappings.

In summary, $\Gamma$ and $\Delta$ offer a comprehensive evaluation of abstract reasoning in LLMs. $\Gamma$ measures basic reasoning accuracy under standard conditions, establishing a performance baseline. $\Delta$, as a diagnostic metric, reveals the nature of this accuracy by quantifying performance drop under symbol remapping. This distinction helps differentiate accuracy from *Rule-Inductive* abstraction (low $\Delta$) versus *Rule-Given* memorization (high $\Delta$). This dual-metric approach is crucial for deeper insights into LLMs' abstract reasoning, moving beyond superficial accuracy evaluations.

### 3.4. Comparison with Existing Abstract Reasoning Benchmarks

Existing benchmarks offer AI reasoning insights, but incompletely assess abstract reasoning per our framework. GSM8K, MATH, BIG-Bench Hard mainly evaluate *Rule-Given Reasoning*, susceptible to memorization and lacking memory dependence measures. ARC emphasizes *Rule-Inductive Reasoning* and reduces memorization via novelty, but its visual format is suboptimal for LLMs and neglects *Rule-Given* abilities. Thus, no existing benchmark fully evaluates both Rule-Given and Rule-Inductive reasoning in a unified framework, nor offers memory dependence metrics. This critical gap—especially distinguishing abstraction from memorization—highlights the need for novel benchmarks like ours and $\Delta$, specifically designed for rigorous, diagnostic evaluation of abstract reasoning in LLMs by directly addressing both reasoning types and memory dependence.

### 3.5. Theoretical Validity of $\Gamma$ and $\Delta$ Metrics in Abstract Reasoning Assessment

While our framework and metrics provide an intuitive approach to evaluating abstract reasoning, we need to establish their theoretical soundness. Just as cognitive science research validates psychological measures through formal analysis, we must demonstrate that our metrics $\Gamma$ and $\Delta$ truly capture the intended aspects of abstract reasoning. Here, we develop three foundational theorems that validate our approach.

Our first theorem addresses a fundamental question: Can we trust $\Gamma$ as a reliable indicator of an LLM's basic reasoning capabilities? This validation is essential before we can build upon it to assess more complex reasoning patterns.

**Theorem 3.7** (Validity of $\Gamma$ for Rule-Given Potential). *Let $\hat{H}$ be an LLM approximating the abstract reasoning function $H = \mathcal{R}e \circ f$. For a sufficiently high threshold $\gamma \in [0, 1]$:*

$$P(\hat{H}(c, r) = q \mid (c, r, q) \in \mathcal{T}) \geq \gamma \tag{7}$$

*where $\mathcal{T}$ is the test set of concrete instances, rules, and expected conclusions.*

This theorem establishes that achieving a high $\Gamma$ score indicates strong Rule-Given reasoning potential in the model.

With the foundation of $\Gamma$'s validity established, we turn to a more nuanced question: How can we ensure that high performance truly reflects abstract understanding rather than memorization? This leads us to our second theorem, which validates $\Delta$ as a measure of genuine rule-inductive reasoning.

**Theorem 3.8** (Validity of $\Delta$ for Rule-Inductive Abstraction). *For sufficiently small $\delta \in [0,1]$ and high $\gamma \in [0,1]$:*

$$\Delta \leq \delta \wedge \Gamma \geq \gamma \qquad (8)$$

This theorem demonstrates that low memory dependence combined with high accuracy indicates true Rule-Inductive abstraction capabilities.

Having established the validity of both metrics individually, we naturally arrive at the question of how to interpret them in combination. Our final theorem provides a formal framework for this unified interpretation.

**Theorem 3.9** (Score Range Interpretation).

$$\mathcal{F}(\Gamma, \Delta) = w_1 \Gamma + w_2 (1 - \Delta) \qquad (9)$$

*where $w_1, w_2 \geq 0$ and $w_1 + w_2 = 1$*

This theorem establishes a continuous mapping $\mathcal{F} : [0,1] \times [0,1] \to [0,1]$ that provides a valid measure of abstract reasoning ability by combining both metrics.

These theorems provide a rigorous mathematical foundation for our evaluation framework, demonstrating that $\Gamma$ and $\Delta$ effectively capture different yet complementary aspects of abstract reasoning ability in LLMs. The proofs for all theorems are presented in Appendix A.2.

# 4. Benchmark Design

We designed a novel symbolic task benchmark to rigorously evaluate abstract reasoning in LLMs, operationalizing our theoretical framework and addressing limitations of existing evaluations. This benchmark provides a direct and robust assessment of abstract reasoning capabilities.

## 4.1. Benchmark Design Principles: Theoretical Grounding

Our benchmark design, theoretically grounded in Section 3, prioritizes effective LLM evaluation of abstract reasoning. We use one-dimensional text input for relevance to LLMs. Systematic symbol remappings are employed to rigorously

test for genuine abstraction, moving beyond mere token memorization. Tasks are theoretically aligned with the definitions of abstraction (f) and reasoning ($\mathcal{Re}$) established in Section 3, and are categorized (BC-SR) for structured analysis. This tiered category system (BC-SR) also ensures scalability across varying levels of cognitive complexity, from basic arithmetic to function inference. Finally, objective diagnosis of abstract reasoning and memory dependence is achieved through our quantitative metrics, $\Gamma$ and $\Delta$ and supported by automated analysis.

## 4.2. Task Categories

Our benchmark tasks are designed with tiered complexity to comprehensively evaluate abstract reasoning. Basic Computation (BC) tasks assess fundamental arithmetic skills in the decimal system, serving as a baseline for computational abilities. Extended Calculation (EC) tasks evaluate the ability to perform diverse computations beyond basic arithmetic, testing the breadth of computational skills. Number Base Reasoning (NBR) tasks rigorously test the generalization of arithmetic reasoning beyond decimal, forcing abstraction of underlying arithmetic principles. Math Application (MA) tasks evaluate multi-step mathematical word problem solving, assessing higher-level reasoning in applied settings, adopting GSM8K style problems for comparison. Symbolic Math Abstraction (SMA) tasks probe inductive mathematical reasoning and abstracting symbolic functions from numerical data, testing pattern discovery. Symbolic Reasoning (SR) tasks directly assess abstract rule application and symbolic manipulation, testing rule identification and application with abstract symbols, independent of domain knowledge.

Detailed task specifications and examples are provided in the appendix.

# 5. Experimental Results and Analysis

In this study, we assessed the abstract reasoning abilities of Large Language Models (LLMs) across different scales and types. Our evaluation encompassed 7B-scale and 70B-scale open-source models, API-accessible models, and agent frameworks (agentchat(autogen), react, llm debate), the latter leveraging `gpt-4o-mini`. For open-source and API models, we employed greedy decoding for inference, while agent frameworks were used with default settings. To evaluate performance, we utilized two prompting strategies for each model type: Direct Prompting (DP) and Chain-of-Thought (CoT), primarily zero-shot CoT, with the exception of Math Application (MA) tasks under CoT, where 8-shot examples based on GSM8K-like data were incorporated. All experiments were conducted on local machines equipped with 8 NVIDIA GPUs, including A800 and 3090 models. For consistent and reliable evaluation of model

*Table 1.* Abstract Reasoning Performance (Γ Scores) of Representative LLMs across Different Scales and Types.

| MODEL | AVERAGE SCORE |
|---|---|
| **7B-SCALE MODELS** | |
| GLM-4-9B-CHAT | 0.17 |
| GLM-4-9B-CHAT COT | 0.25 |
| GEMMA-2-9B-IT | 0.18 |
| GEMMA-2-9B-IT COT | 0.19 |
| LLAMA-3.1-8B-INSTRUCT | 0.17 |
| LLAMA-3.1-8B-INSTRUCT COT | 0.21 |
| QWEN2.5-7B-INSTRUCT | 0.19 |
| QWEN2.5-7B-INSTRUCT COT | 0.34 |
| **32B-SCALE MODELS** | |
| QWQ-32B-PREVIEW | 0.22 |
| QWQ-32B-PREVIEW COT | 0.50 |
| **70B-SCALE MODELS** | |
| LLAMA-3.3-70B-INSTRUCT | 0.22 |
| LLAMA-3.3-70B-INSTRUCT COT | 0.43 |
| QWEN2.5-72B-INSTRUCT | 0.28 |
| QWEN2.5-72B-INSTRUCT COT | 0.47 |
| **API-BASED MODELS** | |
| GPT-4O-MINI | 0.40 |
| GPT-4O-MINI COT | 0.43 |
| GEMINI-2.0-FLASH-EXP | 0.29 |
| GEMINI-2.0-FLASH-EXP COT | 0.52 |
| GEMINI-2.0-FLASH-THINKING-EXP | 0.39 |
| GEMINI-2.0-FLASH-THINKING-EXP COT | 0.54 |
| DEEPSEEK V3 | 0.48 |
| DEEPSEEK V3 COT | 0.41 |
| **AGENTS FRAMEWORKS** | |
| AGENTCHAT(AUTOGEN) | 0.60 |
| REACT | 0.46 |
| LLM DEBATE | 0.35 |

outputs across all tasks, we employed `gpt-4o-mini` to parse responses and determine answer correctness. Based on this comprehensive evaluation framework, our experiments revealed several critical insights into current LLM capabilities and limitations.

Through systematic evaluation focusing on rule-inductive reasoning and generalization beyond memorized patterns, our analysis uncovered several fundamental limitations in current LLMs: (1) widespread failure in non-decimal arithmetic reasoning, with even 70B-scale models showing near-zero performance on number base tasks; (2) strong dependence on specific operand symbols rather than abstract patterns, as quantified by our Memory Dependence Score Δ; and (3) a complex trade-off in Chain-of-Thought prompting, where improved task performance often comes at the cost of increased memory dependence. These findings suggest fundamental limitations in LLMs' ability to perform genuine abstract reasoning, particularly when faced with novel symbolic representations.

### 5.1. Quantitative Results and Analysis

Our evaluation encompasses performance metrics across different model scales (7B-70B), prompting strategies, and

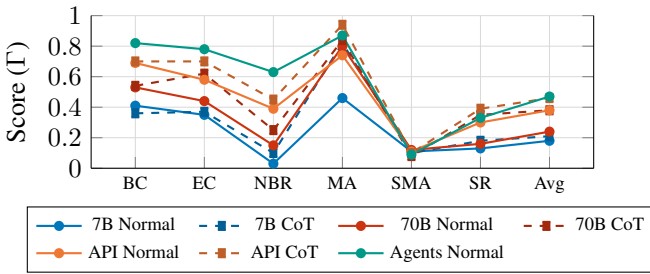

*Figure 3.* Abstract Reasoning Performance across Tasks, Model Types, and Prompting (Normal/CoT).

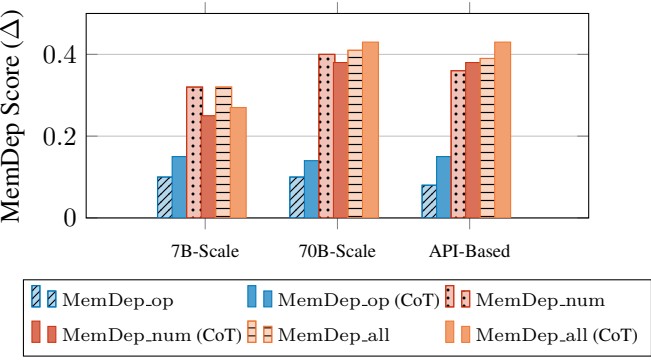

*Figure 4.* Memory Dependence Score (Δ) across Model Types and Prompting Strategies. Operand memory dependence is consistently higher.

task categories. Figure 3 presents a detailed comparison of Abstract Reasoning Score (Γ) across model types and tasks. On decimal-based tasks (Basic Computation (BC), Extended Calculation (EC)), API models achieve average Γ exceeding 0.5, while 7B/70B models reach over 0.4 (Table 3 for detailed scores). However, Number Base Reasoning (NBR) performance collapses across all model scales, with average Γ below 0.1, highlighting a significant abstraction gap in numerical generalization.

Math Application (MA) tasks show substantial gains with Chain-of-Thought (CoT) prompting, particularly for 7B models, suggesting CoT's effectiveness in guiding multi-step, familiar arithmetic problems. In contrast, Symbolic Reasoning (SR) and Symbolic Math Abstraction (SMA) tasks remain challenging, with average Γ generally below 0.3 even for larger models, indicating persistent difficulty in abstract symbolic manipulation and function inference.

The Memory Dependence Score Δ provides crucial insights into the nature of model capabilities. Analysis reveals consistently higher dependence on operand symbols (MemDep_num) compared to operator symbols (MemDep_op) across models and tasks. For instance,

*Table 2.* Memory Dependence Analysis (Δ Scores) of Representative LLMs: Measuring Abstraction vs. Memorization.

| MODEL | OP | NUM | ALL |
|---|---|---|---|
| **7B-SCALE MODELS** | | | |
| GLM-4-9B-CHAT | 0.14 | 0.26 | 0.27 |
| GLM-4-9B-CHAT COT | 0.23 | 0.30 | 0.34 |
| GEMMA-2-9B-IT | 0.10 | 0.37 | 0.37 |
| GEMMA-2-9B-IT COT | 0.33 | 0.36 | 0.40 |
| LLAMA-3.1-8B-INSTRUCT | 0.11 | 0.35 | 0.35 |
| LLAMA-3.1-8B-INSTRUCT COT | 0.14 | 0.23 | 0.23 |
| QWEN2.5-7B-INSTRUCT | 0.10 | 0.36 | 0.37 |
| QWEN2.5-7B-INSTRUCT COT | 0.23 | 0.34 | 0.40 |
| **32B-SCALE MODELS** | | | |
| QWQ-32B-PREVIEW | 0.02 | 0.23 | 0.21 |
| QWQ-32B-PREVIEW COT | 0.31 | 0.50 | 0.58 |
| **70B-SCALE MODELS** | | | |
| LLAMA-3.3-70B-INSTRUCT | 0.11 | 0.42 | 0.43 |
| LLAMA-3.3-70B-INSTRUCT COT | 0.17 | 0.37 | 0.43 |
| QWEN2.5-72B-INSTRUCT | 0.07 | 0.42 | 0.43 |
| QWEN2.5-72B-INSTRUCT COT | 0.12 | 0.41 | 0.47 |
| **API-BASED MODELS** | | | |
| GPT-4O-MINI | 0.11 | 0.30 | 0.35 |
| GPT-4O-MINI COT | 0.20 | 0.47 | 0.44 |
| GEMINI-2.0-FLASH-EXP 0.09 | 0.41 | 0.44 | |
| GEMINI-2.0-FLASH-EXP COT | 0.15 | 0.35 | 0.41 |
| GEMINI-2.0-FLASH-THINKING-EXP | 0.10 | 0.48 | 0.51 |
| GEMINI-2.0-FLASH-THINKING-EXP COT | 0.08 | 0.33 | 0.41 |
| DEEPSEEK V3 | 0.08 | 0.37 | 0.41 |
| DEEPSEEK V3 COT | 0.15 | 0.40 | 0.47 |
| **AGENTS FRAMEWORKS** | | | |
| AGENTCHAT(AUTOGEN) | 0.25 | 0.50 | 0.56 |
| REACT | 0.41 | 0.59 | 0.70 |
| LLM DEBATE | 0.21 | 0.40 | 0.41 |

Llama-3.3-70B-Instruct exhibits $\mathrm{MemDep\_num} = 0.42$ vs. $\mathrm{MemDep\_op} = 0.11$, indicating stronger reliance on specific operand symbols. CoT prompting sometimes increases $\Delta$, as observed in glm-4-9b-chat where $\Delta_{\mathrm{NUM}}$ rises from 0.26 to 0.30 with CoT. Even agent frameworks show high memory dependence, with react reaching $\mathrm{MemDep\_all}$ of 0.70, suggesting persistent token-specific reasoning despite architectural sophistication.

### 5.2. Pattern Analysis and Failure Modes

Our analysis reveals a clear task difficulty hierarchy: SMA(hardest) → NBR → SR → EC, BC → MA (easiest), reflecting fundamental limitations in LLMs' abstract reasoning capabilities. This is particularly evident in NBR tasks, where low $\Gamma$ scores across all model scales demonstrate a critical failure to generalize arithmetic abstraction ($f$) beyond decimal representations, often triggering catastrophic failures or random outputs, especially in smaller models. Similarly, SMA tasks frequently yielded apparently random responses, pointing to a fundamental weakness in inductive function inference – the ability to abstract symbolic rules ($\mathcal{Re}$) from numerical examples. These patterns signify not merely performance limitations, but fundamental deficits in

true rule induction for novel symbolic systems, revealing core cognitive capabilities currently lacking in LLMs.

Chain-of-Thought prompting exhibits nuanced effects across different task categories. While it reduces arithmetic errors in MA and EC tasks, it shows limited benefit for NBR, SMA, or SR tasks. This pattern suggests that CoT primarily aids in structuring procedural steps within familiar domains rather than enhancing fundamental abstraction capabilities. Moreover, the concurrent rise in memory dependence ($\Delta$) with CoT in some cases indicates a potential trade-off: procedural guidance might reinforce token-specific reasoning at the expense of genuine abstraction.

MA tasks, despite their word problem format, proved relatively easier across models, largely attributable to their reliance on familiar decimal arithmetic and extensively trained problem-solving patterns. This observation further supports our finding that models excel in domains with familiar symbolic representations but struggle with novel abstract patterns.

Agent frameworks, despite showing improved accuracy scores, exhibited persistent token memorization patterns in SMA and SR tasks. This suggests that even sophisticated multi-agent architectures fail to overcome fundamental abstraction limitations, instead potentially reinforcing superficial pattern matching through their interaction protocols.

### 5.3. Deeper Analysis: Training Data Impact and Human Baseline

To further probe the nature of LLM abstract reasoning, we conducted supplementary experiments investigating the impact of training data and establishing a human performance baseline. Detailed experimental setups is provided in Appendix A.11.

First, we fine-tuned Llama-3.1-8B-Instruct on datasets with and without our systematic symbol remapping. Fine-tuning on data containing remapped symbols significantly improved performance on tasks with the *same* remapping structure (e.g., accuracy on the "fixed_len_chat_bit_dataset" task increased from 13% to 60% for direct prompting after fine-tuning on remapped data). However, this improvement showed limited generalization; performance on tasks with *different*, unseen remapping structures (e.g., "fixed_len_chat_str_dataset") did not show a similar uplift and, in some cases, even degraded slightly (see Table 4). This suggests that the model tended to memorize specific mapping patterns rather than acquiring a generalizable abstract rule-application capability.

Second, we benchmarked human performance using undergraduate students with a computer science background on a subset of these challenging tasks. Humans demonstrated robust abstract reasoning, achieving near-perfect

accuracy (97%) on the remapped bitwise operation task ("fixed_len_chat_bit_dataset") and strong performance (87%) on non-decimal arithmetic (e.g., "add_base3_raw_dataset"), significantly outperforming the LLMs (Table 4). Even on the more complex remapped string operation task ("fixed_len_chat_str_dataset"), where LLM performance was particularly low, humans achieved 47% accuracy, highlighting a substantial gap.

These findings underscore that while specific training can enhance performance on familiar remapped structures, current LLMs still struggle with genuine generalization in abstract symbolic reasoning and fall considerably short of human capabilities in flexibly applying abstract rules to novel representations.

### 5.4. Implications and Future Directions

The Memory Dependence Score $\Delta$ serves as a critical diagnostic tool, revealing how LLMs rely more heavily on memorizing numerical tokens ($\mathrm{MemDep\_num} > \mathrm{MemDep\_op}$) than learning abstract rules. This operand-specific memorization fundamentally limits models' ability to generalize across different symbolic representations. As Figure 6 demonstrates, higher $\Delta$ directly correlates with reduced generalization capability.

These findings indicate that LLMs' abstract reasoning limitations stem not from scale issues, but from their fundamental reliance on memorized patterns rather than representation-invariant rules. Advanced techniques like multi-agent frameworks and CoT prompting improve performance metrics but fail to address this core abstraction deficit.

We propose three critical research directions: (1) Symbolic data augmentation through systematic permutation to reduce token memorization, (2) Development of functional embedding spaces to promote abstraction beyond lexical forms, and (3) Extension of benchmarks to evaluate generalization across physical and causal reasoning domains.

The $\Delta$ metric provides a quantitative measure for tracking progress toward true symbolic generalization, essential for developing AI systems with genuine abstract reasoning capabilities rather than mere pattern matching.

## 6. Conclusion

We tackled the crucial challenge of rigorously evaluating abstract reasoning in LLMs through a theoretically robust framework. By defining abstract reasoning as the interplay of abstraction and reasoning, we derived validity for our metrics, $\Gamma$ and $\Delta$, and designed a symbol remapping benchmark to compel genuine generalization. Extensive evaluations using this benchmark exposed a critical limitation: current LLMs, despite strengths in familiar domains, exhibit a pro-

found deficit in abstract symbolic reasoning, hindered by significant memory dependence and limited generalization, even with advanced techniques.The benchmark dataset, generation scripts, and evaluation code proposed in this paper will be publicly available at `https://github.com/MAC-AutoML/abstract-reason-benchmark`.

## Acknowledgements

This work was supported by the National Science Fund for Distinguished Young Scholars (No.62025603), the National Natural Science Foundation of China (No. U21B2037, No. U22B2051, No. U23A20383, No. 62176222, No. 62176223, No. 62176226, No. 62072386, No. 62072387, No. 62072389, No. 62002305 and No. 62272401), and the Natural Science Foundation of Fujian Province of China (No. 2021J06003, No.2022J06001).

## Impact Statement

This work introduces novel metrics, $\Gamma$ (Abstract Reasoning Score) and $\Delta$ (Memory Dependence Score), to rigorously benchmark abstract reasoning in Large Language Models (LLMs). The primary positive impact lies in fostering the development of more robust and generalizable AI. By providing a more nuanced evaluation framework that disentangles genuine reasoning from memorization, our benchmark ($\Gamma$) and diagnostic tool ($\Delta$) can guide research towards LLMs capable of deeper understanding and more reliable performance in novel, complex scenarios. This could accelerate progress in AI safety and alignment by promoting models that operate on underlying principles rather than superficial correlations, potentially leading to more predictable and interpretable systems. Improved abstract reasoning capabilities are crucial for advancing AI applications in science, education, and critical problem-solving domains where true generalization is paramount.

Potential negative impacts, common to advancements in AI evaluation, include the risk of "benchmark hacking", where models might be narrowly optimized for $\Gamma$ and $\Delta$ without commensurate gains in holistic cognitive abilities. Furthermore, while our work aims to improve LLM robustness, the development of more capable AI systems inherently carries dual-use concerns and necessitates ongoing ethical scrutiny regarding their deployment and societal consequences. The insights gained from these metrics, if not carefully contextualized, could also be misinterpreted, leading to an overestimation or underestimation of AI capabilities in specific reasoning facets. We encourage the community to use these tools responsibly, as part of a broader suite of evaluations, to cultivate AI that is not only more capable but also more aligned with human values and societal benefit.

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

# A. Appendix

In this appendix, we present comprehensive performance tables of our experimental results, including $\Gamma$ and $\Delta$ metrics across various models and tasks. In addition to these detailed tables, we provide further clarifications on symbol mapping protocol to ensure a complete understanding of our benchmark and evaluation process.

## A.1. Abstract Reasoning in Machine Learning Context

From a *statistical learning* perspective, the composite abstract reasoning function $H = \mathcal{R}e \circ f$ can be viewed as a hypothesis class in machine learning. Here, the abstraction mapping $f$ acts as a feature extractor, transforming concrete inputs into abstract representations, and the reasoning function $\mathcal{R}e$ serves as a decision rule, operating on these abstract features to produce conclusions. In the context of Large Language Models (LLMs), this perspective provides valuable insights. The abstraction mapping $f$ can be loosely associated with the embedding and encoding layers of an LLM, which process input tokens and extract relevant features. The reasoning function $\mathcal{R}e$, on the other hand, aligns with the decoder and prediction components, which utilize the extracted features to generate output sequences.

However, applying this framework to machine learning, especially in the context of evaluating abstract reasoning in LLMs, introduces unique challenges that distinguish it from traditional machine learning approaches:

1. **Symbolic Invariance:** Traditional machine learning data augmentation techniques, like image rotation or cropping, aim to improve robustness to variations in input data. For abstract reasoning, however, we require a more nuanced form of invariance: *symbolic invariance*. This means that a truly abstractly reasoning model should be invariant to symbolic transformations that alter the surface form of inputs (e.g., symbol remapping) but preserve the underlying abstract structure and rules. As illustrated in Figure 5, remapping digits or operators should not fundamentally impair the model's ability to perform the reasoning task if it has genuinely grasped the abstract principles. This is in stark contrast to memorization-based approaches that rely on specific token identities.

2. **Composite Learning:** In many traditional machine learning tasks, feature extraction and decision-making might be implicitly learned or treated as separate stages. In abstract reasoning, however, the emphasis is on learning a *composite process* $H = \mathcal{R}e \circ f$. This means that the model must learn to seamlessly integrate the abstraction mapping $f$ and the reasoning function $\mathcal{R}e$. It's not sufficient to just have a good feature extractor or a good rule applier in isolation; the model must learn to systematically unify these components to effectively perform abstract reasoning across different rules and tasks. This composite nature requires a more holistic learning approach compared to simply optimizing individual modules.

3. **Information Compression through Abstraction:** Beyond standard generalization metrics like accuracy, abstract reasoning highlights the critical role of *information compression through abstraction*. A hallmark of abstract reasoning is the ability to distill complex, concrete instances into concise, abstract features. This compression is not merely about reducing dimensionality; it's about selectively discarding irrelevant surface details while preserving the essential, reasoning-relevant structure. An effective abstract reasoning system should operate primarily on these compressed abstract representations, rather than directly on the high-dimensional raw input space. The efficiency and generalizability of abstract reasoning are intrinsically linked to this ability to achieve meaningful information compression.

4. **Rule Availability during Training (Rule-Given vs. Rule-Inductive):** The nature of abstract reasoning tasks, in relation to an LLM's training data, significantly impacts the difficulty and the type of reasoning required. *Rule-given abstract reasoning* tasks are those where the underlying rules or patterns are likely to have been encountered, implicitly or explicitly, during the LLM's extensive pre-training. In such cases, the model can potentially leverage its vast learned knowledge and memory to perform the task, effectively recalling and applying pre-existing associations. For example, tasks involving decimal arithmetic, common sense reasoning based on everyday knowledge, or applying frequently seen symbolic transformations might fall into this category. The model's performance could be enhanced by its ability to access and utilize relevant information acquired during training.

   Conversely, *rule-inductive abstract reasoning* tasks present a much greater challenge. These tasks involve novel rules or patterns that are unlikely to have been directly or frequently encountered in the training data. To succeed, the LLM must go beyond simple recall and pattern matching; it must genuinely induce the underlying rule from the provided task description or examples and then apply this newly learned rule to solve the problem. Tasks involving arithmetic in unfamiliar number bases, inferring novel symbolic operations from a few examples, or applying abstract rules defined

using novel symbols are characteristic of rule-inductive reasoning. These tasks demand a higher level of abstraction, requiring the model to actively learn and generalize rules on-the-fly, rather than relying on pre-existing knowledge. Therefore, rule-inductive tasks are crucial for rigorously evaluating the true abstract reasoning capacity of LLMs, as they minimize reliance on memorization and maximize the need for genuine, flexible rule learning and application.

This naturally raises a crucial question for evaluating machine learning models, especially LLMs: how can we effectively measure whether a model has truly learned abstract reasoning, particularly rule-inductive abstract reasoning, rather than merely memorizing patterns or exploiting superficial cues? Consider again a model solving binary arithmetic. It might achieve correct results either through a genuine understanding of abstract arithmetic rules applicable to any base, or by simply memorizing common input-output patterns specific to binary operations represented with '0' and '1'. Traditional accuracy metrics alone, while useful, are insufficient to distinguish between these fundamentally different approaches. We need evaluation methodologies that can probe deeper into the nature of a model's reasoning and reveal whether it is grounded in genuine abstraction or mere memorization.

### A.2. Proofs of Theorems

#### A.2.1. PROOF OF THEOREM 3.7: VALIDITY OF $\Gamma$ FOR RULE-GIVEN POTENTIAL

Let $\gamma$ be a sufficiently high threshold and consider model $\hat{H}$ with Abstract Reasoning Score $\Gamma \geq \gamma$.

Step 1: By definition of $\Gamma$, we have:

$$\frac{1}{N} \sum_{i=1}^{N} \nVdash [\hat{H}(c_i, r_i) = q_i] \geq \gamma \tag{10}$$

Step 2: For large sample size $N$, by the Law of Large Numbers:

$$P(\hat{H}(c, r) = q | (c, r, q) \in \mathcal{T}) \geq \gamma \tag{11}$$

Step 3: Let $f$ be the abstraction mapping and $\mathcal{R}e$ the reasoning function. For high $\gamma$:

$$P(\mathcal{R}e(f(c), r) = q) \geq \gamma \tag{12}$$

Therefore, high accuracy on original symbolic representations implies Rule-Given proficiency.

#### A.2.2. PROOF OF THEOREM 3.8: VALIDITY OF $\Delta$ FOR RULE-INDUCTIVE ABSTRACTION

Consider model $\hat{H}$ with $\Delta \leq \delta$ and $\Gamma \geq \gamma$.

Step 1: By definition of $\Delta$:

$$|\Gamma - \Gamma_M| \leq \delta \tag{13}$$

where $\Gamma_M$ is accuracy under symbol mapping $M$.

Step 2: For any concrete instance $c \in \mathcal{C}$:

$$|P(\hat{H}(c, r) = q) - P(\hat{H}(M(c), r) = M(q))| \leq \delta \tag{14}$$

Step 3: Given $\Gamma \geq \gamma$, we have:

$$P(\hat{H}(c, r) = q) \geq \gamma \ P(\hat{H}(M(c), r) = M(q)) \qquad \geq \gamma - \delta \tag{15}$$

Step 4: Let $f$ be the abstraction mapping. For small $\delta$:

$$f(c) \approx f(M(c)) \implies \mathcal{R}e(f(c), r) \approx \mathcal{R}e(f(M(c)), r) \tag{16}$$

This invariance to symbol mapping demonstrates Rule-Inductive abstraction.

A.2.3. PROOF OF THEOREM 3.9: SCORE RANGE INTERPRETATION

We construct and validate the mapping $\mathcal{F}$.

Step 1: Define $\mathcal{F}$ as weighted sum:

$$\mathcal{F}(\Gamma, \Delta) = w_1 \Gamma + w_2(1 - \Delta) \tag{17}$$

Step 2: Verify properties for extreme cases:

Case 1 (High Ability): When $\Gamma \approx 1$ and $\Delta \approx 0$:

$$\mathcal{F}(\Gamma, \Delta) \approx w_1 + w_2 = 1 \tag{18}$$

Case 2 (Medium Ability): When $\Gamma \approx 0.5$ and $\Delta \approx 0.5$:

$$\mathcal{F}(\Gamma, \Delta) \approx 0.5 \tag{19}$$

Case 3 (Low Ability): When $\Gamma \approx 0$ or $\Delta \approx 1$:

$$\mathcal{F}(\Gamma, \Delta) \approx 0 \tag{20}$$

Step 3: $\mathcal{F}$ is continuous and monotonic in $\Gamma$ and $-\Delta$, ensuring smooth transitions between ability levels.

Therefore, $\mathcal{F}$ provides a valid mapping from scores to abstract reasoning ability.

## A.3. Model Performance and Memory Dependence Table

## A.4. Diagrams

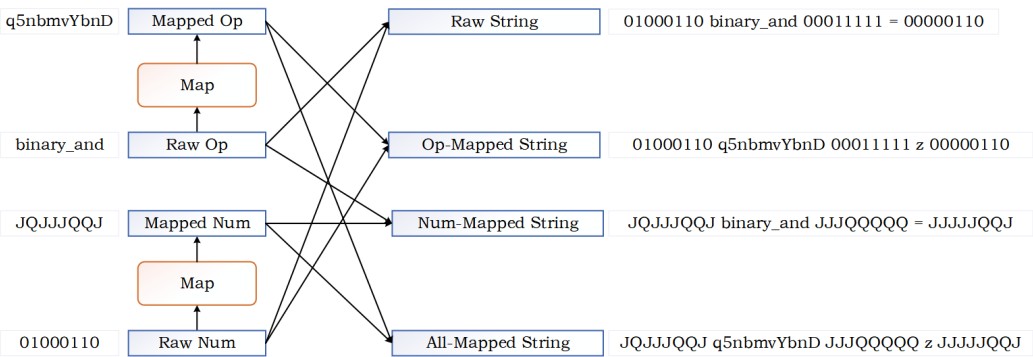

*Figure 5.* Illustration of the symbolic mapping process for operands and operators.

## A.5. Symbol Mapping Protocols and Implementation

Symbol mapping is central to disentangling abstract reasoning from superficial memorization, forcing focus on rule structures. We employ operand, operator, and combined remapping, as illustrated in Figure 5. To specifically assess memory dependence, Symbolic Reasoning (SR) tasks, derived from Extended Calculation (EC) tasks through these mappings, are used in conjunction with EC to calculate the Memory Dependence Score ($\Delta$). Performance evaluation under remappings quantifies memory dependence ($\Delta$) and the robustness of abstract reasoning. The generation of remapped datasets follows a systematic procedure:

1. **Candidate Symbol Pool Generation:** A pool of candidate symbols for remapping is curated from the Llama-2 7B tokenizer's vocabulary. This vocabulary is filtered to retain only single-character alphanumeric tokens (e.g., 'a'-'z', 'A'-'Z', '0'-'9' that are treated as individual tokens). This ensures that the new symbols are basic, unlikely to carry strong pre-existing semantic meaning in complex sequences, and are distinct from structural elements like spaces or newlines. Let this set of candidate remapping tokens be denoted as $S_{\text{cand}}$.

*Table 3.* Model Performance and Memory Dependence Evaluation. $\Gamma$ values represent accuracy on task categories: Basic Computation (BC), Extended Calculation (EC), Number Base Reasoning (NBR), Math Application (MA), Symbolic Math Abstraction (SMA), Symbolic Reasoning (SR). $\Delta$ values show performance drops under memory dependence tests.

| MODEL | BC | EC | NBR | MA | SMA | SR | AVG | OP | NUM | ALL |
|---|---|---|---|---|---|---|---|---|---|---|
| **7B-SCALE MODELS** | | | | | | | | | | |
| INTERNLM2_5-7B-CHAT (CAI ET AL., 2024) | 0.51 | 0.30 | 0.01 | 0.27 | 0.11 | 0.11 | 0.16 | 0.08 | 0.28 | 0.29 |
| INTERNLM2_5-7B-CHAT COT | 0.40 | 0.39 | 0.02 | 0.82 | 0.13 | 0.23 | 0.24 | 0.10 | 0.25 | 0.27 |
| GLM-4-9B-CHAT (GLM ET AL., 2024) | 0.20 | 0.34 | 0.00 | 0.18 | 0.14 | 0.15 | 0.17 | 0.14 | 0.26 | 0.27 |
| GLM-4-9B-CHAT COT | 0.24 | 0.46 | 0.10 | 0.84 | 0.14 | 0.22 | 0.25 | 0.23 | 0.30 | 0.34 |
| YI-1.5-9B-CHAT-16K (YOUNG ET AL., 2024) | 0.53 | 0.38 | 0.03 | 0.29 | 0.11 | 0.14 | 0.19 | 0.09 | 0.36 | 0.38 |
| YI-1.5-9B-CHAT-16K COT | 0.49 | 0.48 | 0.14 | 0.80 | 0.08 | 0.28 | 0.30 | 0.14 | 0.31 | 0.32 |
| GEMMA-2-9B-IT (TEAM ET AL., 2024) | 0.44 | 0.38 | 0.03 | 0.21 | 0.12 | 0.14 | 0.18 | 0.10 | 0.37 | 0.37 |
| GEMMA-2-9B-IT COT | 0.47 | 0.41 | 0.19 | 0.87 | 0.00 | 0.09 | 0.19 | 0.33 | 0.36 | 0.40 |
| MARCO-O1 (ZHAO ET AL., 2024) | 0.49 | 0.39 | 0.04 | 0.31 | 0.11 | 0.14 | 0.19 | 0.10 | 0.37 | 0.38 |
| MARCO-O1 COT | 0.47 | 0.38 | 0.03 | 0.50 | 0.11 | 0.14 | 0.19 | 0.13 | 0.34 | 0.35 |
| LLAMA-3.1-8B-INSTRUCT (DUBEY ET AL., 2024) | 0.35 | 0.35 | 0.02 | 0.75 | 0.11 | 0.12 | 0.17 | 0.11 | 0.35 | 0.35 |
| LLAMA-3.1-8B-INSTRUCT COT | 0.27 | 0.35 | 0.07 | 0.87 | 0.19 | 0.18 | 0.21 | 0.14 | 0.23 | 0.23 |
| OPENMATH2-LLAMA3.1-8B (TOSHNIWAL ET AL., 2024) | 0.24 | 0.33 | 0.03 | 0.89 | 0.14 | 0.18 | 0.19 | 0.05 | 0.26 | 0.25 |
| OPENMATH2-LLAMA3.1-8B COT | 0.14 | 0.21 | 0.05 | 0.83 | 0.11 | 0.16 | 0.16 | 0.02 | 0.08 | 0.10 |
| QWEN2.5-7B-INSTRUCT (YANG ET AL., 2024) | 0.50 | 0.39 | 0.06 | 0.31 | 0.11 | 0.14 | 0.19 | 0.10 | 0.36 | 0.37 |
| QWEN2.5-7B-INSTRUCT COT | 0.57 | 0.55 | 0.31 | 0.91 | 0.13 | 0.27 | 0.34 | 0.23 | 0.34 | 0.40 |
| QWEN2.5-MATH-7B-INSTRUCT | 0.49 | 0.34 | 0.09 | 0.95 | 0.12 | 0.12 | 0.18 | 0.15 | 0.27 | 0.27 |
| QWEN2.5-MATH-7B-INSTRUCT COT | 0.19 | 0.11 | 0.02 | 0.94 | 0.01 | 0.05 | 0.07 | 0.04 | 0.06 | 0.08 |
| NORMAL AVG | 0.41 | 0.35 | 0.03 | 0.46 | 0.11 | 0.13 | 0.18 | 0.10 | 0.32 | 0.32 |
| COT AVG | 0.36 | 0.37 | 0.10 | 0.82 | 0.10 | 0.18 | 0.21 | 0.15 | 0.25 | 0.27 |
| **32B-SCALE MODELS** | | | | | | | | | | |
| QWQ-32B-PREVIEW (TEAM, 2025) | 0.42 | 0.33 | 0.17 | 0.95 | 0.16 | 0.17 | 0.22 | 0.02 | 0.23 | 0.21 |
| QWQ-32B-PREVIEW COT | 0.78 | 0.83 | 0.53 | 0.95 | 0.13 | 0.40 | 0.50 | 0.31 | 0.50 | 0.58 |
| **70B-SCALE MODELS** | | | | | | | | | | |
| LLAMA-3.3-70B-INSTRUCT | 0.52 | 0.44 | 0.09 | 0.84 | 0.11 | 0.16 | 0.22 | 0.11 | 0.42 | 0.43 |
| LLAMA-3.3-70B-INSTRUCT COT | 0.53 | 0.69 | 0.21 | 0.96 | 0.09 | 0.43 | 0.43 | 0.17 | 0.37 | 0.43 |
| LLAMA-3.1-NEMOTRON-70B-INSTRUCT | 0.49 | 0.42 | 0.10 | 0.74 | 0.13 | 0.17 | 0.23 | 0.11 | 0.39 | 0.38 |
| LLAMA-3.1-NEMOTRON-70B-INSTRUCT COT | 0.45 | 0.66 | 0.19 | 0.95 | 0.12 | 0.38 | 0.40 | 0.15 | 0.42 | 0.45 |
| OPENMATH2-LLAMA3.1-70B | 0.47 | 0.43 | 0.12 | 0.94 | 0.13 | 0.15 | 0.23 | 0.13 | 0.40 | 0.41 |
| OPENMATH2-LLAMA3.1-70B COT | 0.48 | 0.42 | 0.24 | 0.53 | 0.04 | 0.19 | 0.25 | 0.12 | 0.35 | 0.37 |
| QWEN2.5-72B-INSTRUCT | 0.64 | 0.49 | 0.29 | 0.68 | 0.11 | 0.18 | 0.28 | 0.07 | 0.42 | 0.43 |
| QWEN2.5-72B-INSTRUCT COT | 0.71 | 0.72 | 0.39 | 0.95 | 0.08 | 0.43 | 0.47 | 0.12 | 0.41 | 0.47 |
| NORMAL AVG | 0.53 | 0.44 | 0.15 | 0.80 | 0.12 | 0.16 | 0.24 | 0.10 | 0.40 | 0.41 |
| COT AVG | 0.54 | 0.62 | 0.25 | 0.84 | 0.08 | 0.35 | 0.38 | 0.14 | 0.38 | 0.43 |
| **API-BASED MODELS** | | | | | | | | | | |
| GPT-4O-MINI | 0.54 | 0.63 | 0.05 | 0.90 | 0.12 | 0.44 | 0.40 | 0.11 | 0.30 | 0.35 |
| GPT-4O-MINI COT | 0.57 | 0.72 | 0.25 | 0.92 | 0.10 | 0.41 | 0.43 | 0.20 | 0.47 | 0.44 |
| GEMINI-1.5-FLASH (TEAM ET AL., 2023) | 0.59 | 0.50 | 0.28 | 0.36 | 0.04 | 0.32 | 0.34 | 0.06 | 0.28 | 0.27 |
| GEMINI-1.5-FLASH COT | 0.61 | 0.64 | 0.27 | 0.93 | 0.02 | 0.37 | 0.40 | 0.18 | 0.36 | 0.42 |
| GEMINI-2.0-FLASH-EXP | 0.72 | 0.52 | 0.26 | 0.56 | 0.14 | 0.19 | 0.29 | 0.09 | 0.41 | 0.44 |
| GEMINI-2.0-FLASH-EXP COT | 0.75 | 0.77 | 0.47 | 0.94 | 0.13 | 0.48 | 0.52 | 0.15 | 0.35 | 0.41 |
| GEMINI-2.0-FLASH-THINKING-EXP | 0.80 | 0.60 | 0.76 | 0.95 | 0.12 | 0.20 | 0.39 | 0.10 | 0.48 | 0.51 |
| GEMINI-2.0-FLASH-THINKING-EXP COT | 0.81 | 0.72 | 0.77 | 0.95 | 0.13 | 0.43 | 0.54 | 0.08 | 0.33 | 0.41 |
| DEEPSEEK V3 (LIU ET AL., 2024A) | 0.80 | 0.67 | 0.64 | 0.96 | 0.13 | 0.37 | 0.48 | 0.08 | 0.37 | 0.41 |
| DEEPSEEK V3 COT | 0.79 | 0.66 | 0.50 | 0.96 | 0.12 | 0.29 | 0.41 | 0.15 | 0.40 | 0.47 |
| NORMAL AVG | 0.69 | 0.58 | 0.39 | 0.74 | 0.11 | 0.30 | 0.38 | 0.08 | 0.36 | 0.39 |
| COT AVG | 0.70 | 0.70 | 0.45 | 0.94 | 0.10 | 0.39 | 0.46 | 0.15 | 0.38 | 0.43 |
| **AGENTS FRAMEWORKS** | | | | | | | | | | |
| AGENTCHAT(AUTOGEN) (WU ET AL., 2023) | 0.96 | 0.88 | 0.95 | 0.90 | 0.10 | 0.43 | 0.60 | 0.25 | 0.50 | 0.56 |
| REACT (YAO ET AL., 2023) | 0.97 | 0.83 | 0.76 | 0.85 | 0.06 | 0.25 | 0.46 | 0.41 | 0.59 | 0.70 |
| LLM DEBATE (DU ET AL., 2023) | 0.53 | 0.63 | 0.19 | 0.86 | 0.12 | 0.31 | 0.35 | 0.21 | 0.40 | 0.41 |
| NORMAL AVG | 0.82 | 0.78 | 0.63 | 0.87 | 0.09 | 0.33 | 0.47 | 0.29 | 0.50 | 0.56 |

2. **Identification of Original Symbols for a Task Instance:** For each original task instance, consisting of few-shot examples and a question-answer pair, we first identify the set of all unique non-whitespace characters, $U_{\text{orig}}$, present in its textual representation (both questions and answers within the examples and the target QA pair).

3. **Random Bijective Symbol Mapping Establishment:** A random bijective (one-to-one and onto) mapping function, $M_{\text{sym}} : U_{\text{orig}} \rightarrow S'_{\text{cand}}$, is established. Here, $S'_{\text{cand}}$ is a randomly selected subset of $S_{\text{cand}}$ such that $|S'_{\text{cand}}| = |U_{\text{orig}}|$. This ensures that each unique original symbol is mapped to a unique novel candidate symbol.

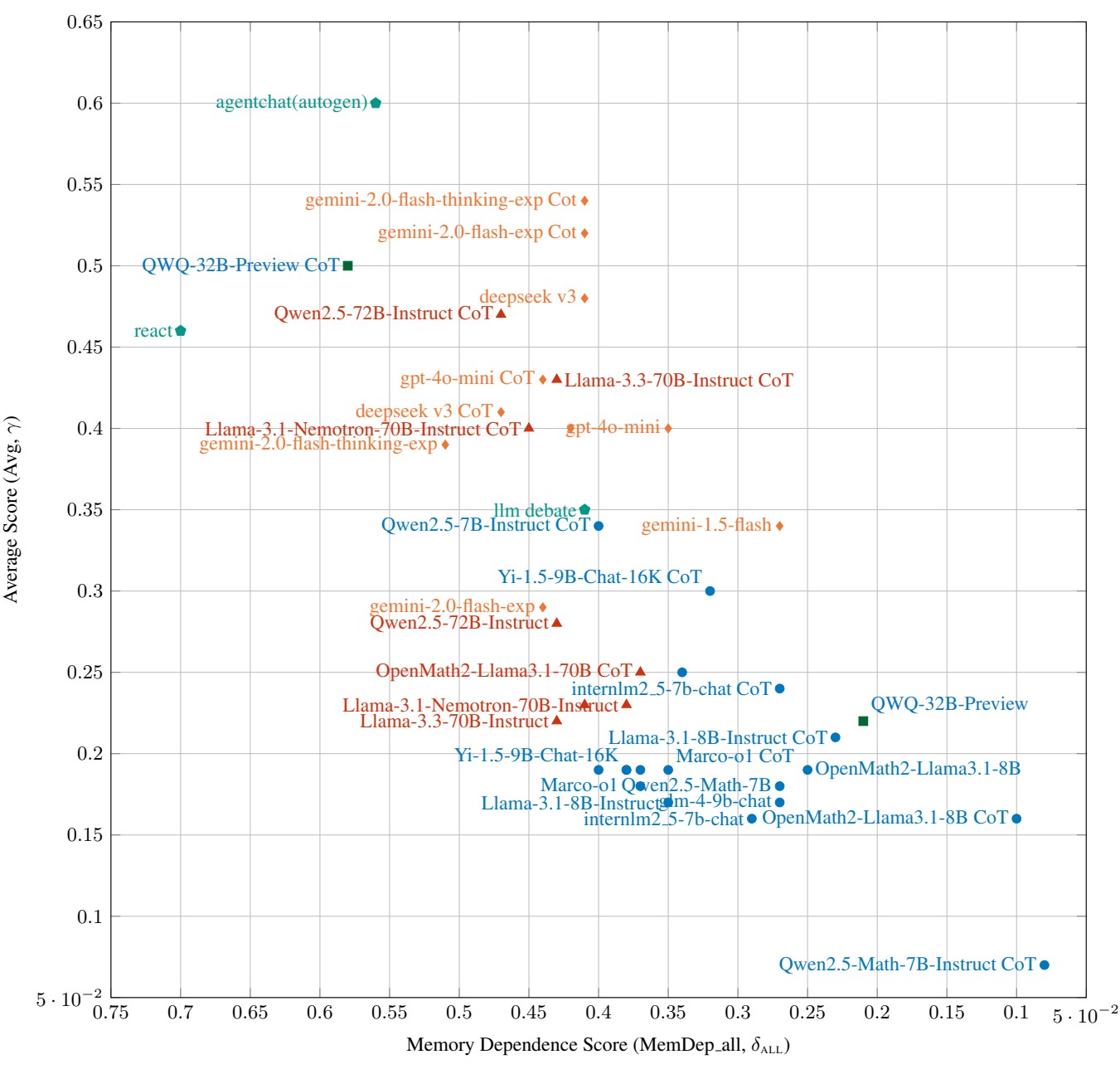

*Figure 6.* Memory Dependence vs. Average Score(Large Scale)

4. **Generation of Remapped Task Instances under Different Strategies:** The original task instances are then transformed by applying character-wise substitution based on specific mapping strategies to produce remapped instances. The core strategies are:

*Table 4.* Detailed Performance Comparison: Fine-tuned Llama-3.1-8B-Instruct Variants and Human Baseline. Accuracies are percentages. (*) denotes training on unmapped symbols; (**) denotes training on fully remapped symbols. "Untrained" is the base model. "N/A" indicates data not applicable for that row/column (e.g., human baseline for fine-tuning, or models not subjected to fine-tuning on these specific configurations).

| DATASET | MODEL / PARTICIPANT GROUP | UNTRAINED | FINE-TUNED (EPOCH 8) | HUMAN BASELINE |
|---|---|---|---|---|
| FIXED_LEN_CHAT_BIT_DATASET | LLAMA-3.1-8B-INSTRUCT* DIRECT | 13% | 18% | 97% |
| | LLAMA-3.1-8B-INSTRUCT* CoT | 5% | 7% | |
| | LLAMA-3.1-8B-INSTRUCT** DIRECT | 13% | 60% | |
| | LLAMA-3.1-8B-INSTRUCT** CoT | 5% | 11% | |
| | GEMINI-2.0-FLASH-THINKING-EXP DIRECT | 57% | N/A | |
| | GEMINI-2.0-FLASH-THINKING-EXP CoT | 46% | N/A | |
| FIXED_LEN_CHAT_STR_DATASET | LLAMA-3.1-8B-INSTRUCT* DIRECT | 27% | 27% | 47% |
| | LLAMA-3.1-8B-INSTRUCT* CoT | 3% | 7% | |
| | LLAMA-3.1-8B-INSTRUCT** DIRECT | 27% | 25% | |
| | LLAMA-3.1-8B-INSTRUCT** CoT | 3% | 2% | |
| | GEMINI-2.0-FLASH-THINKING-EXP DIRECT | 19% | N/A | |
| | GEMINI-2.0-FLASH-THINKING-EXP CoT | 17% | N/A | |
| ADD_BASE3_RAW_DATASET | GEMINI-2.0-FLASH-THINKING-EXP | 75% | N/A | 87% |

- **Raw (No Remapping):** The original task instance is used as-is. This serves as the baseline for calculating $\Gamma$.
- **Combined Remapping (All Symbols):** The mapping $M_{\text{sym}}$ (derived from all unique characters $U_{\text{orig}}$) is applied to all characters in the task instance's examples and QA pair. This generates the fully remapped dataset variant used for one type of $\Delta$ calculation.
- **Operand-Specific Remapping (Number/Value Symbols):** First, unique characters constituting only the operands (e.g., '0', '1' in bit strings; digits in numerical operations) are identified from $U_{\text{orig}}$, let this set be $U_{\text{operand}}$. A distinct random bijective mapping $M_{\text{operand}} : U_{\text{operand}} \rightarrow S''_{\text{cand}}$ is created. This mapping is then applied *only* to the operand characters within the task instance. Operator symbols remain in their original form.
- **Operator-Specific Remapping (Operation Symbols):** Similarly, unique characters representing operators (e.g., 'binary and', '+', or their symbolic representations in the examples) are identified, let this set be $U_{\text{operator}}$. A distinct random bijective mapping $M_{\text{operator}} : U_{\text{operator}} \rightarrow S'''_{\text{cand}}$ is created. This mapping is applied *only* to the operator characters. Operand symbols remain unchanged.

For all remapping strategies, the underlying problem structure and the rules implied by the examples are preserved; only the surface symbolic representation is altered.

5. **Dataset Assembly:** By applying these strategies, we generate multiple versions of each sub-dataset: a raw version, a fully remapped version, an operand-remapped version, and an operator-remapped version. These variants allow for the calculation of $\Gamma$ (from raw performance) and different facets of $\Delta$ by comparing performance on raw versus remapped datasets.

This systematic remapping allows us to quantify how much a model's performance relies on specific familiar tokens versus its ability to generalize to abstract patterns represented by novel symbols.

### A.6. Dataset Details

Our benchmark comprises 82 sub-datasets, categorized across six main task categories: Basic Computation (BC), Extended Calculation (EC), Number Base Reasoning (NBR), Math Application (MA), Symbolic Math Abstraction (SMA), and Symbolic Reasoning (SR). With the exception of the Math Application (MA) category which utilizes the GSM8K dataset, each sub-dataset contains 96 examples, resulting in a total of 9095 samples across the entire benchmark. During dataset construction, we ensured diversity in task instructions and samples, and meticulously excluded potential duplicate phrasings to prevent models from solving tasks via superficial pattern matching. The naming convention for sub-datasets indicates key variations: prefixes like 'var_len' or 'fixed_len' denote variable or fixed operand lengths, respectively; 'chat' signifies a conversational prompt format. The core of the filename often indicates the specific operation type (e.g., 'bit' for bitwise operations, 'str' for string manipulations, 'strop' for specific string operations). Furthermore, suffixes such as 'raw' identify tasks using original, unmapped symbols, while 'op_map', 'num_map', and 'all_map' specify whether symbol remapping is

applied to operators, operands (numbers), or both, respectively. A detailed list of sub-datasets within each task category is provided below:

### A.6.1. BASIC COMPUTATION (BC) DATASETS

```
chat_add_dataset
chat_div_dataset
chat_sub_dataset
chat_mul_dataset
```

### A.6.2. EXTENDED CALCULATION (EC) DATASETS

```
var_len_chat_list_cnt_raw_dataset
var_len_chat_strop_raw_dataset
fixed_len_chat_substr_raw_dataset
fixed_len_chat_bit_raw_dataset
fixed_len_chat_str_raw_dataset
var_len_chat_bitop_raw_dataset
var_len_chat_str_raw_dataset
var_len_chat_bit_shift_raw_dataset
var_len_chat_list_raw_dataset
fixed_len_chat_bitop_raw_dataset
chat_square_dataset
var_len_chat_bit_raw_dataset
var_len_chat_data_raw_dataset
var_len_chat_set_raw_dataset
fixed_len_chat_strop_raw_dataset
fixed_len_chat_bit_shift_raw_dataset
```

### A.6.3. MATH APPLICATION (MA) DATASETS

```
dataset_gsm8k (1319 samples)
```

### A.6.4. NUMBER BASE REASONING (NBR) DATASETS

```
chat_add_base3_raw_dataset
chat_add_base4_raw_dataset
chat_base5_raw_dataset
chat_sub_base4_raw_dataset
chat_mul_base3_raw_dataset
chat_base3_raw_dataset
chat_mul_base4_raw_dataset
chat_base4_raw_dataset
chat_add_base5_raw_dataset
chat_sub_base3_raw_dataset
chat_mul_base5_raw_dataset
chat_sub_base5_raw_dataset
```

### A.6.5. SYMBOLIC MATH ABSTRACTION (SMA) DATASETS

```
chat_quadratic_dataset
chat_triangle_wave_dataset
chat_sawtooth_wave_dataset
chat_square_wave_dataset
chat_cosine_dataset
chat_linear_dataset
```

```
chat_sine_dataset
```

## A.6.6. SYMBOLIC REASONING (SR) DATASETS

```
var_len_chat_bitop_num_dataset
fixed_len_chat_str_op_dataset
fixed_len_chat_bit_op_dataset
var_len_chat_list_cnt_num_dataset
var_len_chat_bit_dataset
fixed_len_chat_bit_shift_dataset
var_len_chat_bit_shift_num_dataset
var_len_chat_bit_shift_dataset
var_len_chat_bit_op_dataset
var_len_chat_list_op_dataset
var_len_chat_list_num_dataset
var_len_chat_set_dataset
var_len_chat_list_cnt_dataset
var_len_chat_str_num_dataset
var_len_chat_bitop_op_dataset
var_len_chat_list_cnt_op_dataset
fixed_len_chat_bit_shift_op_dataset
fixed_len_chat_substr_num_dataset
var_len_chat_bitop_dataset
fixed_len_chat_str_num_dataset
var_len_chat_strop_dataset
var_len_chat_str_op_dataset
var_len_chat_list_dataset
fixed_len_chat_bitop_num_dataset
fixed_len_chat_str_dataset
var_len_chat_bit_shift_op_dataset
var_len_chat_set_num_dataset
fixed_len_chat_bitop_dataset
var_len_chat_strop_op_dataset
var_len_chat_bit_num_dataset
fixed_len_chat_substr_dataset
fixed_len_chat_strop_dataset
fixed_len_chat_substr_op_dataset
fixed_len_chat_strop_num_dataset
fixed_len_chat_bit_shift_num_dataset
fixed_len_chat_strop_op_dataset
var_len_chat_strop_num_dataset
fixed_len_chat_bit_dataset
fixed_len_chat_bit_num_dataset
var_len_chat_set_op_dataset
fixed_len_chat_bitop_op_dataset
var_len_chat_str_dataset
fixed_len_chat_substr_op_dataset
```

**Total Samples:** 9095

**Dataset Structure:** Each dataset consists of input-output pairs designed to evaluate specific abstract reasoning skills as detailed in Section 4. The input and output formats are consistent across all sub-datasets within each task category, ensuring a standardized evaluation framework. The 'Avg' column in Table 3 represents the average of the Abstract Reasoning Score ($\Gamma$) across all sub-datasets within the corresponding task category.

## A.7. Example Demonstrations

Here we provide example prompts and instances from our benchmark to illustrate the task format and the symbol mapping methodology.

### A.7.1. EXAMPLE FROM EXTENDED CALCULATION (EC) CATEGORY

**Prompt:** The task is to identify patterns and discover rules from the provided examples, then answer a question. The symbols in the question may not have their usual meanings, so carefully analyze the rules and expressions before providing your final answer in the format: "Answer: The answer is {your answer}."

**Examples:**

- Question: 01000110 binary_and 00011111 =

- Answer: The answer is 00000110.

- Question: 00011100 binary_and 00010001 =

- Answer: The answer is 00010000.

- Question: 01011110 binary_and 00001101 =

- Answer: The answer is 00001100.

- ...

**Question:** 00100111 binary_and 01100111 = **Answer:** The answer is 00100111.

### A.7.2. EXAMPLE FROM SYMBOLIC REASONING (SR) CATEGORY WITH SYMBOL MAPPING

**Prompt:** The task is to identify patterns and discover rules from the provided examples, then answer a question. The symbols in the question may not have their usual meanings, so carefully analyze the rules and expressions before providing your final answer in the format: "Answer: The answer is {your answer}."

**Examples:**

- Question: JQJJJQQJ q5nbmvYbnD JJJQQQQQ z

- Answer: The answer is JJJJJQQJ.

- Question: JJJQQQJJ q5nbmvYbnD JJJQJJJQ z

- Answer: The answer is JJJQJJJJ.

- Question: JQJQQQQJ q5nbmvYbnD JJJJQQJQ z

- Answer: The answer is JJJJQQJJ.

- ...

**Question:** JJQJJQQQ q5nbmvYbnD JQQJJQQQ z **Answer:** The answer is JJQJJQQQ.

## A.8. Example Instances for Task Categories

Here we provide example instances for each task category in our benchmark, drawing from the benchmark document provided:

- **BC (Basic Computation):**
  - Addition: '27 + 15817 = ?'

- – Subtraction: '100 - 25 = ?'
- – Multiplication: '12 * 8 = ?'
- – Division: '24 / 8 = ?'

- **EC (Extended Calculation):** This category includes a variety of extended computational tasks:

  - – **Square Root Calculation:** 'sqrt(625) = ?'
  - – **Bitwise Operations:**
    - * Binary AND: '01000110 binary_and 00011111 = ?'
    - * Binary OR: '01000110 binary_or 00011111 = ?'
    - * Binary NOT: 'binary_not 01010101 = ?'
  - – **Bit Shift Operations:**
    - * Logical Left Shift: '00000110 bit_shift_left 2 = ?'
    - * Logical Right Shift: '00000110 bit_shift_right 2 = ?'
    - * Circular Right Shift: '00000110 circular_right_shift 1 = ?'
  - – **Bit Manipulation Operations:**
    - * Check Bit: '01000110 check_bit 1 = ?' (Check if bit at position 1 is set)
    - * Set Bit: '01100010 set_bit 0 = ?' (Set bit at position 0 to 1)
    - * Toggle Bit: '00010100 toggle_bit 6 = ?' (Flip bit at position 6)
  - – **String Manipulation:**
    - * Reverse String: 'reverse('algorithm') = ?'
    - * Concatenate Strings: 'concatenate('hello', 'world') = ?'
    - * Repeat String: 'repeat('go', 3) = ?'
    - * Get String Length: 'get_length('benchmarking') = ?'
    - * Substring Containment (in order): 'ebhbgfhdbcfgfbhbbegaafaaceechhfhadacdabb contains(in order) bffd = ?'
  - – **Set Operations:**
    - * Difference: 'difference(0, 2, 4, 0, 1, 4) = ?'
    - * Union: 'union(a, b, c, c, d, e) = ?'
    - * Intersection: 'intersection(1, 2, 3, 3, 4, 5) = ?'
  - – **List Operations:**
    - * Sort List: 'sort([5, 2, 8, 1, 9]) = ?'
    - * Filter List: 'filter([1, 5, 10, 3, 8], 5) = ?' (Filter elements greater than 5)
    - * Deduplicate List: 'deduplicate([a, b, a, c, c, b, d]) = ?'
    - * Maximum Value in List: 'max([12, 5, 23, 8, 15]) = ?'
    - * Minimum Value in List: 'min([12, 5, 23, 8, 15]) = ?'
    - * Median Value in List: 'median([3, 1, 4, 1, 5, 9, 2, 6]) = ?'
    - * Mode Value in List: 'mode([a, b, c, b, a, a, d]) = ?'
  - – **Date Calculation:**
    - * Days Between Dates: 'days_between_dates([2024, 07, 29], [2021, 10, 31]) = ?'

- **NBR (Number Base Reasoning):**

  - – Ternary Addition: '2200102 (base3) + 11100111 (base3) = ? (base3)'
  - – Quaternary Subtraction: '321 (base4) - 13 (base4) = ? (base4)'
  - – Quinary Multiplication: '23 (base5) * 4 (base5) = ? (base5)'
  - – Base Conversion: '25 (base10) to base 3 = ? (base3)'

- **MA (Math Application):**

  - – GSM8K style problems, for example: "If Maria buys 3 apples and each apple costs \$0.50, and she also buys a banana for \$1.00, how much does she spend in total?"

- **SMA (Symbolic Math Abstraction):**

  – Infer function type and parameters from input-output pairs:
    * Linear Function: Input-Output pairs: (1, 5), (2, 8), (3, 11). Question: fun(4) = ? (Function is f(x) = 3x + 2)
    * Quadratic Function: Input-Output pairs: (1, 2), (2, 7), (3, 14). Question: fun(4) = ? (Function is f(x) = x$\hat{2}$ + 1)
    * Exponential Function: Input-Output pairs: (1, 6), (2, 18), (3, 54). Question: fun(4) = ? (Function is f(x) = 2 * 3$\hat{x}$)
    * Logarithmic Function: Input-Output pairs: (1, 0), (e, a), (e$\hat{2}$, 2a). Question: fun(e$\hat{3}$) = ? (Function is f(x) = a * ln(x))
    * Sine Function: Input-Output pairs: (0, 0), ($\pi$/2, a), ($\pi$, 0). Question: fun(3$\pi$/2) = ? (Function is f(x) = a * sin(x))

- **SR (Symbolic Reasoning):**

  – For the Symbolic Reasoning (SR) dataset, we utilized tasks from the Extended Calculation (EC) category, excluding tasks involving square root calculations (sqrt) and date-related operations (data).

## A.9. Benchmark Task Generation Details

**Definition A.1** (Task Generation Function). For each task category $c$, the generation function $G_c$ is defined as:

$$G_c : \mathcal{R} \times \mathcal{P} \to \{(x_i, y_i)\}_{i=1}^n$$

where:

- $\mathcal{R}$ is the rule space.

- $\mathcal{P}$ is the parameter space for task configuration.

- $(x_i, y_i)$ are input-output pairs satisfying $y_i = R(f(x_i), r)$ for $r \in \mathcal{R}$.

## A.10. Illustrative Case Studies: Success and Failure in Abstract Reasoning

To further illustrate the performance characteristics of LLMs on our benchmark, this section presents two specific case studies: one failure and one success. For brevity, the few-shot examples provided in the full prompt are omitted here; we focus on the final question posed, the ground truth, the model's full 'raw output' (which includes its Chain-of-Thought process), and an analysis of its reasoning.

### A.10.1. FAILURE CASE: NON-DECIMAL ARITHMETIC (BASE-5 ADDITION)

**Task Context:** The model was tasked with performing addition in base-5, with explicit hints and examples indicating the base of operation. This tests the model's ability to generalize arithmetic principles to non-decimal systems.

- **Model:** DeepSeek-V3 (with CoT prompting)

- **Dataset:** `chat_add_base5_raw_dataset`

- **Question Posed (final part of prompt):** `\nHint:  This is base 5 operation.\nQuestion:  411 + 421200 =\n\n` (Examples omitted)

- **Ground Truth Answer:** `422111` (base 5)

- **Model's Extracted Final Answer:** `422211` (base 5)

**Model's Full 'raw output':**

```
To solve \( 411 + 421200 \) in base 5, we follow these steps:

### Step 1: Align the numbers by their place values
We align the numbers so that the digits are in the correct place value columns:
```

```
   421200
+     411
-------
```

### Step 2: Add the digits column by column,
starting from the rightmost column (units place)
We add the digits in each column, keeping in mind that this is base 5,
so any sum greater than or equal to 5 will carry over to the next column.

1. **Rightmost column (units place):**
   \( 0 + 1 = 1 \).
   Write down **1**.

2. **Next column (fives place):**
   \( 0 + 1 = 1 \).
   Write down **1**.

3. **Next column (twenty-fives place):**
   \( 2 + 4 = 6 \).
   Since 6 is greater than or equal to 5,
   we subtract 5 and carry over 1 to the next column.
   Write down **1** and carry over **1**.

4. **Next column (one hundred twenty-fives place):**
   \( 1 + 0 = 1 \), plus the carry-over of 1 gives \( 1 + 1 = 2 \).
   Write down **2**.

5. **Next column (six hundred twenty-fives place):**
   \( 2 + 0 = 2 \).
   Write down **2**.

6. **Next column (three thousand one hundred twenty-fives place):**
   \( 4 + 0 = 4 \).
   Write down **4**.

7. **Leftmost column (fifteen thousand six hundred twenty-fives place):**
   \( 4 + 0 = 4 \).
   Write down **4**.

### Step 3: Write the final result
Combining all the digits from left to right, the result is:

**422211**.

### Final Answer:
The answer is **422211**.

**Analysis of Model's Reasoning:** The model's 'raw output' indicates an attempt to follow a base-5 addition procedure. It correctly identifies the need for carrying when a column sum is $\geq 5$. For instance, in "Step 2, part 3," it calculates $2 + 4 = 6$, which becomes '1' with a carry of '1' in base 5. The critical error occurs in the handling of place values for numbers of differing lengths. While '421200' (base 5) is a six-digit number and '411' (base 5) is a three-digit number, the model's "Step 2, part 7" incorrectly processes the most significant digit of '421200' (the leftmost '4') as if it were a seventh digit to be added to an implicit leading zero from the shorter number. This leads to an erroneous extra '4' at the beginning of its

sum, resulting in '422211' instead of the correct '422111'. This case demonstrates a failure in accurately abstracting and applying place value concepts in a non-decimal system, even when the fundamental carry logic is partially understood. The CoT, while outlining steps, did not prevent this structural misunderstanding.

A.10.2. SUCCESS CASE: SYMBOLIC REASONING WITH REMAPPING (BITWISE AND)

**Task Context:** The model was presented with few-shot examples where two 8-character strings composed of novel symbols ('Z' and '3') were combined with a distractor operator string to produce an 8-character output string. The task required inferring both the meaning of the novel symbols and the underlying logical operation.

- **Model:** Gemini-2.0-Flash-Thinking-Exp (with CoT prompting)

- **Dataset:** `fixed_len_chat_bit_dataset` (a fully remapped bitwise operation task)

- **Question Posed (final part of prompt):** `Question:  ZZ333ZZ3 LHWFRwgFWr ZZZ3ZZ3Z G\n\nLet's think step by step.  \n\n` (Examples omitted)

- **Ground Truth Answer:** `ZZZ3ZZZZ`

- **Model's Extracted Final Answer:** `ZZZ3ZZZZ`

**Model's Full 'raw output':**

```
Answer: The answer is ZZZ3ZZZZ.

**Step-by-step reasoning:**
1. **Convert 'Z' and '3' to binary digits:** Treat 'Z' as 0 and '3' as 1.
   - First string: ZZ333ZZ3 becomes 00111001.
   - Third string: ZZZ3ZZ3Z becomes 00010010.

2. **Perform bitwise AND operation:**
Apply the bitwise AND operation to each corresponding bit of
the binary representations of the first and third strings.

   00111001 (First string)
   AND
   00010010 (Third string)
   ---------
   00010000 (Result)

   - Position 1: 0 AND 0 = 0
   - Position 2: 0 AND 0 = 0
   - Position 3: 1 AND 0 = 0
   - Position 4: 1 AND 1 = 1
   - Position 5: 1 AND 0 = 0
   - Position 6: 0 AND 0 = 0
   - Position 7: 0 AND 1 = 0
   - Position 8: 1 AND 0 = 0

3. **Convert the binary result back to 'Z' and '3':**
Convert the binary result 00010000 back to a string of 'Z's and '3's,
where 0 is 'Z' and 1 is '3'.
   - 00010000 becomes ZZZ3ZZZZ.

4. **Compare with the provided answer:**
```

```
The derived answer ZZZ3ZZZZ matches the provided answer
for the question ZZ333ZZ3 LHWFRwgFWr ZZZ3ZZ3Z G.

Final Answer: The final answer is $\boxed{ZZZ3ZZZZ}$
```

**Analysis of Model's Reasoning:** The model's 'raw output' clearly articulates a successful multi-step abstract reasoning process:

1. **Symbolic Abstraction:** In "Step 1," the model correctly infers the semantic mapping of the novel symbols: 'Z' to binary '0' and '3' to binary '1'. It then accurately converts the two relevant input strings ('ZZ333ZZ3' and 'ZZZ3ZZ3Z') into their binary representations ('00111001' and '00010010', respectively). The distractor elements 'LHWFRwgFWr' and 'G' are correctly ignored.

2. **Operational Inference and Execution:** In "Step 2," the model correctly identifies the underlying operation as a bitwise AND applied to the two abstracted binary strings. It then performs this operation flawlessly, yielding the binary result '00010000'.

3. **Reverse Symbolic Mapping:** In "Step 3," the model converts the binary result back to the original symbolic domain, correctly translating '00010000' to 'ZZZ3ZZZZ'.

This case exemplifies successful abstract reasoning. The model was not merely pattern matching surface tokens but demonstrated an understanding of the underlying logical structure by (a) mapping novel symbols to a known representational system (binary), (b) inferring the correct logical operation from examples, and (c) applying this inferred rule to new inputs. The CoT output provides transparent evidence of this robust reasoning process.

### A.11. Supplementary Experimental Details

This section provides further details on the Large Language Model (LLM) evaluation setup, the fine-tuning experiments, and the human baseline evaluation.

#### A.11.1. LLM EVALUATION SETUP

The main LLM evaluations were conducted utilizing two distinct server configurations, selected based on model scale and computational needs: one server equipped with 8 NVIDIA GeForce RTX 3090 GPUs (24GB VRAM each), and another with 8 NVIDIA A800 GPUs (80GB VRAM each). For inference, we generally employed a generation configuration with a temperature of 1e-7 and a maximum of 2096 new tokens. The inference batch size was adjusted according to available GPU memory and model size, prioritizing model-specific default generation parameters where applicable.

#### A.11.2. FINE-TUNING SETUP FOR LLAMA-3.1-8B-INSTRUCT

To investigate the impact of training data on abstract reasoning with remapped symbols, we fine-tuned the Llama-3.1-8B-Instruct model. Input prompts and their corresponding answers were formatted as a continuous sequence, specifically structured as `[INST] {prompt} [/INST] {answer}`. We utilized the AutoTokenizer from the "meta-llama/Llama-3.1-8B-Instruct" pretrained model, configuring it with padding on the left side and setting the pad token to be the tokenizer's end-of-sequence (EOS) token. All inputs were tokenized to a maximum length of 2048, with truncation and padding applied as necessary.

Two distinct training data configurations were employed. The "Unmapped Symbols Training (*)" involved fine-tuning the model on 2,000 samples generated using original, unmapped symbols, structured similarly to our "fixed_len_chat_bit_raw_dataset". Conversely, the "Fully Mapped Symbols Training (**)" fine-tuned the model on 2,000 samples where symbols (both operands and operators) were systematically remapped, mirroring the structure of our "fixed_len_chat_bit_dataset".

Key hyperparameters, based on our experimental script, included a per-device batch size of 4, 2 gradient accumulation steps, the Paged AdamW 8-bit optimizer, a learning rate of 2e-5, and a cosine learning rate scheduler with a warmup ratio of 0.03. BF16 precision was used, the seed was set to 42, and gradient checkpointing was enabled.

Post fine-tuning, the performance of these models was assessed on two datasets: the "fixed_len_chat_bit_dataset" (where the remapping structure was encountered during the 'Fully Mapped' training phase) and the "fixed_len_chat_str_dataset" (which presented an unseen remapping structure and task type), to evaluate both learning and generalization. The Chain-of-Thought (CoT) variants mentioned in our earlier rebuttal refer to the application of CoT prompting strategies during the inference phase with these already fine-tuned models; the fine-tuning data itself did not explicitly include CoT examples. Detailed results, encompassing pre-fine-tuning (untrained) baselines and post-fine-tuning performance for both direct prompting and CoT variants, are collated in Table 4.

### A.11.3. HUMAN BASELINE EVALUATION PROTOCOL

To establish a human performance baseline for comparative analysis with LLMs, we recruited four undergraduate students, each with a background in computer science. These participants were tasked with a subset of 16 samples from each of the following datasets: the "fixed_len_chat_bit_dataset" (for remapped bitwise operations), the "fixed_len_chat_str_dataset" (for remapped string operations), and the "add_base3_raw_dataset" (for non-decimal arithmetic, specifically base-3 addition with original symbols).

During the evaluation, participants received the identical problem descriptions and examples that were provided to the LLMs in their respective evaluation settings. They were instructed to infer the underlying rules from these provided examples and subsequently solve the posed questions. It is important to note that no explicit training or instruction regarding the specific symbol mappings was given to the participants beyond what could be deduced from the in-prompt examples. Performance was quantified by calculating accuracy based on the correctness of their final answers. The aggregated human performance across these tasks is detailed and compared with LLM performance in Table 4.

