# OpenReview forum: "Benchmarking Abstract and Reasoning Abilities Through A Theoretical Perspective"
_ICML.cc/2025/Conference — ICML 2025 poster_

### Official Review · Reviewer_SZxB · 2025-02-26

**Overall Recommendation:** 4

**Summary:**

This paper introduces a rigorously designed and theoretically grounded benchmark for a crucial yet underexplored area: the evaluation of abstract reasoning in Large Language Models (LLMs). The work establishes a clear mathematical framework that defines abstract reasoning as the ability to extract invariant patterns and consistently apply rules, independent of superficial representations. Central to the contribution are two novel, complementary metrics: the Abstract Reasoning Score (Γ) to quantify baseline accuracy and the Memory Dependence Score (Δ), a diagnostic metric innovatively designed to reveal the degree to which models rely on memorization versus genuine abstraction. Through systematic symbol remapping in carefully designed rule-based tasks, the benchmark compels models to demonstrate true pattern recognition. Extensive empirical evaluations across a diverse suite of LLMs, encompassing varying scales and prompting strategies, convincingly demonstrate the limitations of current models in abstract reasoning, particularly in symbolic manipulation and generalization. The paper's findings powerfully advocate for the use of this benchmark to guide future progress in AI systems capable of robust abstract thought, representing a significant step forward in LLM evaluation.

**Claims And Evidence:**

The paper’s core claim, asserting that contemporary LLMs exhibit significant deficits in abstract reasoning and are demonstrably prone to memorization—especially when confronted with novel symbolic representations—is convincingly and thoroughly substantiated. The experimental results are particularly compelling: the stark performance decline observed in Number Base Reasoning (NBR) tasks, even for the largest models, alongside the insightful diagnostic capabilities of the Memory Dependence Score (Δ), provide robust empirical support. For example, the consistent increase in Δ observed across models highlights a clear reliance on specific symbol tokens. The benchmark's innovative design, based on systematic symbol remapping within rule-based tasks, is demonstrably effective in isolating and measuring abstract reasoning, moving beyond superficial pattern matching or simple statistical correlations. The comprehensive evaluations, spanning a wide array of models, from 7B to 70B scale, and diverse prompting techniques, further reinforce the strength of the evidence presented. The paper effectively and persuasively argues against the sufficiency of relying solely on standard accuracy metrics for a nuanced assessment of genuine reasoning capacity in LLMs.

**Essential References Not Discussed:**

While the literature review is demonstrably thorough and highly relevant to contemporary LLM research, one could argue for the inclusion of more foundational key works from the pre-LLM era in the field of symbolic AI and rule-based reasoning. For instance, seminal works on production systems, knowledge representation, and early symbolic reasoning architectures, while not strictly essential for grasping the paper’s core contributions in the context of modern LLM evaluation, could provide a more complete and historically nuanced perspective on the evolution of AI reasoning research. However, the current literature review is highly focused and effectively addresses the most pertinent prior work directly relevant to the paper's contributions within the current landscape of LLM research.

**Experimental Designs Or Analyses:**

The experimental design is not only well-executed but also remarkably comprehensive in its scope and depth. The authors have systematically and meticulously evaluated a diverse range of LLMs, thoughtfully spanning different model scales (from 7B to 70B), varying architectures, and distinct access methods (encompassing both open-source and API-based models). The deliberate inclusion of different prompting strategies, namely Direct Prompting and Chain-of-Thought (CoT), allows for a nuanced and granular analysis of model behavior and reasoning capabilities under varying conditions and input formats. The detailed and transparent reporting of results across all defined task categories, and particularly the fine-grained analysis of the Memory Dependence Score (Δ) at the level of both operand and operator symbols, provides exceptionally valuable and granular insights into the inherent limitations of current LLMs in abstract reasoning. The automated evaluation process, which incorporates gpt-40-mini for parsing responses and ensuring objective answer correctness assessment, further enhances the objectivity, reliability, and overall rigor of the study.

**Methods And Evaluation Criteria:**

The methodology employed in this work stands as a significant strength. The benchmark design itself, particularly the ingenious incorporation of systematic symbol remapping, represents a valuable methodological contribution to the field of LLM evaluation. The proposed metrics, Γ and Δ, offer a well-defined and crucially, complementary framework for rigorous evaluation. The Memory Dependence Score (Δ) is particularly noteworthy as it provides a diagnostic measure capable of differentiating between genuine abstract reasoning and mere memorization – a vital distinction frequently overlooked in many LLM evaluations. The deliberate use of rule-based tasks, thoughtfully categorized into Basic Computation (BC), Extended Calculation (EC), Number Base Reasoning (NBR), Math Application (MA), Symbolic Math Abstraction (SMA), and Symbolic Reasoning (SR), provides a structured and scalable approach to evaluating diverse facets of abstract reasoning. The experimental setup, meticulously encompassing a diverse range of models, prompting strategies (including both Direct Prompting and Chain-of-Thought), and computational resources, demonstrates appropriate rigor and careful consideration of experimental controls.

**Other Comments Or Suggestions:**

- Future Benchmark Development and Maintenance: To maximize long-term impact, consider explicitly emphasizing the ongoing development, community engagement, and potential expansions of the benchmark in future publications and releases. Establishing a clear roadmap for incorporating increasingly complex reasoning tasks and adapting to advancements in LLM technology will be crucial for maintaining its continued relevance and utility to the research community.

- Elaborate on Rule-Inductive Reasoning Challenges: Further elaboration, perhaps incorporating more concrete and illustrative examples, specifically detailing the nuanced challenges LLMs demonstrably face when engaging in rule-inductive reasoning tasks, would be highly beneficial for readers less familiar with the intricacies of this cognitive process.

- Dataset Accessibility, Documentation, and Community Building: Ensuring open and easily accessible access to the meticulously curated dataset, coupled with providing exceptionally thorough and user-friendly documentation of its structure, generation process, and intended usage, is absolutely crucial for maximizing reproducibility, fostering broader community adoption, and ensuring the long-term impact of this valuable benchmark. Consider creating a dedicated project website or repository to facilitate access and community contributions.

**Other Strengths And Weaknesses:**

Strengths:

- Novel and Theoretically Grounded Benchmark: The paper introduces a genuinely novel and exceptionally well-theoretically motivated benchmark that directly addresses a critical gap in the rigorous evaluation of abstract reasoning capabilities in LLMs.

- Diagnostic Memory Dependence Score (Δ): The innovative introduction of the Memory Dependence Score (Δ) stands out as a significant strength, providing a uniquely valuable and diagnostic tool for disentangling genuine abstraction from mere memorization.

- Comprehensive and Rigorous Empirical Evaluation: The empirical evaluation is remarkably comprehensive, covering a wide and diverse range of models and prompting strategies, providing exceptionally robust and generalizable results.

- Clear Identification of Critical Research Gaps: The paper effectively and persuasively identifies fundamental limitations in current LLMs' abstract reasoning capabilities and clearly points towards valuable and impactful future research directions, advancing the field.

- Well-Defined Task Categories and Dataset: The structured categorization of tasks (BC, EC, NBR, MA, SMA, SR) and the creation of a well-documented dataset contribute significantly to the reproducibility and usability of the benchmark.

Weaknesses:

- Potential for Further Task Complexity: While already comprehensive, the scope of the benchmark tasks could be further expanded in future iterations to incorporate even more complex and nuanced reasoning scenarios, particularly those involving multi-step inference and more intricate rule structures, to maintain its leading-edge utility as LLMs continue to advance.

- Depth of Rule-Inductive Reasoning Exploration: While the benchmark addresses rule-inductive reasoning, further and deeper exploration of model performance specifically in the domain of rule induction from genuinely novel data – data designed to minimize any reliance on pre-existing patterns or biases learned during pre-training

– could potentially provide even richer and more granular insights into this challenging aspect of abstract reasoning for LLMs.

**Questions For Authors:**

- Benchmark Evolution Roadmap: Could the authors outline their specific plans and roadmap for the continued evolution and expansion of the benchmark? Specifically, what types of increasingly complex reasoning tasks are envisioned for future iterations, and how will the benchmark be adapted to remain effective and challenging as LLMs continue to advance in capabilities? Understanding this long-term vision would further solidify the benchmark's value as a lasting contribution.

- Deep Dive into Rule Induction Failure Modes: To enhance practical understanding for researchers seeking to improve LLM rule induction, could the authors provide more specific, illustrative examples and a more detailed analysis of the typical failure modes observed in LLMs when tackling rule induction tasks? Focusing on concrete examples, particularly from the SMA and SR task categories, would offer valuable insights into the precise nature of the limitations and guide targeted improvement efforts.

- Dataset Access and Community Engagement Strategy: What is the authors' strategy for ensuring broad and sustainable access to the benchmark dataset and fostering community engagement around its use and potential expansion? Clearly articulating plans for dataset accessibility, comprehensive documentation, and community contribution mechanisms would maximize the impact and utility of this valuable resource for the wider AI research community.

**Relation To Broader Scientific Literature:**

This paper is exceptionally well-contextualized within the broader and highly relevant scientific literature concerning abstract reasoning in both cognitive science and artificial intelligence. The related work section is particularly effective in its detailed discussion of existing benchmarks designed to assess AI reasoning, clearly articulating their limitations in the context of rigorously evaluating abstract reasoning within LLMs, and, crucially, in effectively differentiating between genuine abstraction and superficial memorization. The paper thoughtfully builds upon established theoretical perspectives from cognitive science and clearly and persuasively articulates the specific advancements and novel contributions offered by the proposed benchmark and evaluation metrics. The grounding of the work in established cognitive science literature, particularly regarding the distinction between Rule-Given and Rule-Inductive reasoning, is entirely appropriate and significantly strengthens the paper's scholarly context and theoretical underpinnings.

**Theoretical Claims:**

The paper's rigorously developed theoretical framework represents a substantial strength and a defining characteristic of this work. It provides precise formal definitions for key concepts including abstraction mapping, reasoning functions, and composite reasoning functions, offering a solid and mathematically sound conceptual grounding for the benchmark. The theorems presented (3.7, 3.8, 3.9) are not merely assertions but are theoretically validated, demonstrating the validity of Γ for assessing Rule-Given potential and, importantly, Δ as a robust measure for evaluating Rule-Inductive Abstraction capabilities. This level of theoretical rigor significantly distinguishes the paper from purely empirical evaluations and provides a robust foundation for both the empirical findings and the proposed evaluation methodology, enhancing the paper’s overall impact and credibility.

---

> ### Author Rebuttal · Authors · 2025-04-01
>
> ### 1. Task Complexity Expansion
>
> We agree that further expanding task complexity is essential. In future work, we will:
>
> - Introduce multi-step reasoning tasks that require chaining hypotheses, intermediate conclusions, and final integration.
> - Incorporate hierarchical rules with nested, conditional, and conflict resolution components.
> - Implement dynamic difficulty adjustment using parameters (e.g., reasoning steps, rule density, interference factors) to generate tasks that evolve with model proficiency.
> ---
> ### 2. Enhancing Rule-Inductive Reasoning
>
> We plan to deepen our investigation into rule induction by:
>
> - Increasing test difficulty—e.g., training with 8-bit binary data and testing with 16-bit numbers—to challenge the model’s generalization.
> - Recording and validating the reasoning steps, not just the final answer; automated tools will assess the plausibility of the reasoning chain.
> - Including detailed error case analyses, primarily in the Symbolic Math Abstraction (SMA) and Symbolic Reasoning (SR) categories, to illuminate common failure modes.
> ---
> ### 3. Benchmark Evolution and Maintenance
>
> We have a three-stage plan to keep the benchmark state‑of‑the‑art:
>
> - **Short-term:** Open-source the evaluation framework with detailed examples and error analysis.
> - **Mid-term:** Develop a dynamic task difficulty system through parameterization of reasoning steps and rule complexity.
> - **Long-term:** Build an adaptive challenge mechanism that auto-generates advanced tasks as models improve. We will also create robust feedback channels and community platforms to encourage continuous improvement.
> ---
> ### 4. Detailed Examples and Error Analysis for Rule Induction
>
> We will add:
>
> - Inclusion of Detailed Case Studies: We will include a dedicated section in our open-source GitHub repository that presents detailed case studies, particularly focusing on SMA and SR tasks.
> - Error Pattern Analysis: Along with concrete examples, we will provide analyses of common error patterns, explaining the underlying causes and typical failure modes in rule-induced tasks.
>   We thank you for emphasizing the need for more specific examples and error analyses. We agree that these enhancements will provide immediate, valuable insights into model limitations and help guide future improvements.
> ---
> ### 5. Dataset Openness and Community Engagement
>
> To maximize community impact, we will:
>
> - Open-source the complete project on GitHub, including data generation code, symbol mapping, the full dataset, and all evaluation scripts.
> - Provide thorough documentation and user guides.
>
> ---
> We appreciate your detailed feedback. Your recommendations have pinpointed vital areas for improvement. In response, we have developed clear and actionable measures to refine task complexity, enhance rule induction analysis, and ensure the benchmark remains dynamic and accessible. These improvements will help us advance robust abstract reasoning evaluation in LLMs.
>
> Sincerely,
> The Author Team

---

### Official Review · Reviewer_CDZ6 · 2025-03-11

**Overall Recommendation:** 1

**Summary:**

This work build a benchmark of arithmetic computation tasks that targets at the abstract reasoning abilities of large language models, and finds out that the power existing large language models relies on the task domains.

**Claims And Evidence:**

See below

**Essential References Not Discussed:**

N/A

**Experimental Designs Or Analyses:**

The benchmark consists of basic arithmetic tasks, where the areas where abstract reasoning are more broadly discussed (e.g. more complex logical reasoning, commonsense reasoning) is not discussed.

**Methods And Evaluation Criteria:**

The benchmark provides signal that demonstrates the limitation of the reasoning abilities of existing large language models, but not sufficient in supporting the abstract reasoning abilities.

**Other Comments Or Suggestions:**

N/A

**Other Strengths And Weaknesses:**

There are several places worth mentioning in Section 3.
- For Definition 3.1, the key difference between concrete instances and abstract features is not explained. In other words, what makes a string concrete instances instead of abstract features. What is the difference between the mapping C->A and A->C?
- For Definition 3.2, the 'x' operation in 'AxR' is not explained. Meanwhile, the example is confusing as the conclusion applied here is 'This dog is likely to bark' without introducing the origin of the uncertainty here. Does this also apply to all the reasoning functions?
- Theorem 3.7 is hard to understand, the foundation of the validity is not explained.
- For theorem 3.9, the score range interpretation can be irrelevant to \gamma is w1=w2 according to the definition.

**Questions For Authors:**

N/A

**Relation To Broader Scientific Literature:**

The topic would raise interest to the broader scope of audience.

**Theoretical Claims:**

The theorems are very brief and not placed in a well-described framework with all the concepts well explained.

---

> ### Author Rebuttal · Authors · 2025-04-01
>
> ## 1. Concrete Instances vs. Abstract Features
>
> Our paper distinguishes concrete instances (C)—detailed input strings containing surface-level information—from abstract features (A), which capture only the essential properties required for reasoning. For example, a concrete description of a dog may include breed and color, whereas its abstract feature reduces these details to “dog.” Our mapping f: C->A is designed to force models to look beyond token memorization; an inverse mapping (from A->C) is unnecessary for our task.
>
> ---
>
> ## 2. Notation and Example in Definition 3.2
>
> ### (a) Notation "$A \times R$"
>
> In Definition 3.2, the notation $A \times R$ indicates the set of all ordered pairs `(a, r)`, where each `a` belongs to the abstract feature set `A` (Section 3.1) and each `r` belongs to the rule set `R` (Section 3.2). The reasoning function `Re` is defined as
> Re: $A \times R$→Q,
> which means that for any input pair `(a, r)`, the function produces a corresponding conclusion `q` in `Q`. This formulation follows standard mathematical conventions (e.g., see [Cormen et al., 2009]) for combining elements from two previously defined sets.
>
> ### (b) The Example Involving Uncertainty
>
> The phrase “This dog is likely to bark” illustrates that typical human reasoning can involve uncertainty. However, in strictly symbolic or arithmetic tasks, the reasoning outcome is deterministic. The example is meant only to show how abstract features and rules combine, with the particular uncertainty reflecting context rather than a general property of all reasoning functions.
>
> ---
>
> ## 3. Theoretical Foundation of Theorem 3.7
>
> Theorem 3.7 states that if a large language model achieves an abstract reasoning score Γ on our test set T that meets or exceeds a threshold γ, then for every task (c, r, q) in T the probability of correctly outputting q (via the abstraction mapping f and the reasoning function Re) is at least γ. In essence, a high Γ confirms the model’s robust ability to perform abstract reasoning across T.
>
> This theorem is grounded in our rigorous definitions of f and Re, which provide a clear correspondence among the inputs, the applied rules, and the correct outputs. Although the main text offers only a brief proof, more detailed proofs and explanations are available in the Appendix.
>
> ---
>
> ## 4. Score Range Interpretation (Theorem 3.9)
>
> For Theorem 3.9, we define the combined score as
> F(Γ, Δ) = w₁·Γ + w₂·(1 − Δ)
> and, according to Theorem 3.8, a higher threshold γ implies a higher Γ. The weights w1 and w2 are chosen based on task requirements. Even when w1 = w2, the two components remain distinct: Γ measures the reasoning accuracy, while (1 − Δ) reflects the model's robustness against dependence on specific tokens. Thus, the score range interpretation stays relevant as each component captures a distinct aspect of performance.
>
> ---
>
> ## Additional Responses
>
> ### 1. On Supporting Abstract Reasoning Abilities
>
> Our task design is not limited to simple arithmetic operations. Instead, we adopt a symbol remapping strategy that forces models to abandon reliance on surface-level patterns and requires them to extract and apply the underlying abstract rules. For example, in a date calculation task (e.g., `days between dates([2024, 07, 29], [2021, 10, 31]) = ?`), the model is required not only to perform numerical computations but also to understand the intrinsic logic between time and dates, which manifests a higher demand for abstract reasoning. Even large-scale models did not achieve above 60% accuracy on these tasks, which shows that our benchmark is sufficiently challenging for current models’ abstract reasoning abilities.
>
> ### 2. On the Conciseness of Theorems and Theoretical Framework
>
> We acknowledge that our current exposition of definitions and theorems is brief. The definitions of the abstraction mapping (f) and the reasoning function (Re) serve as the foundational building blocks of our mathematical model, and the corresponding theorems (e.g., Theorem 3.7 and Theorem 3.9) follow standard analytical approaches. Detailed proofs and clarifications are provided in the Appendix.
>
> ### 3. On the Benchmark’s Focus on Arithmetic Tasks
>
> While our benchmark primarily features arithmetic and related symbolic tasks, it is not confined to simple calculations. In addition to basic arithmetic, our benchmark incorporates extended calculation tasks, operations in various number bases, and date calculations. Most tasks are designed using symbol remapping techniques to destabilize simple memorization, thereby revealing models’ inability to generalize abstract rules beyond familiar forms.
>
> ---
>
> Because your review was relatively brief, we hope you will raise more questions or concerns to help us further improve the quality of our paper. Your additional feedback will be invaluable as we continue to refine our theoretical framework and experimental design.
>
> Thank you for your review.
>
> Sincerely,
> The Author Team

---

### Official Review · Reviewer_GRws · 2025-03-14

**Overall Recommendation:** 3

**Summary:**

This paper presents a theoretically grounded benchmark to evaluate abstract reasoning in Large Language Models (LLMs). It defines abstract reasoning as extracting essential patterns and applying consistent rules to these patterns. Two metrics, Γ (Abstract Reasoning Score) and ∆ (Memory Dependence Score), are introduced to measure reasoning ability and distinguish abstraction from memorization. Evaluations across various LLMs reveal limitations in non-decimal arithmetic and symbolic reasoning, as well as significant memory dependence. The findings highlight the need for improved abstract reasoning capabilities in LLMs.

**Claims And Evidence:**

The claims are well-supported by the evidence presented. The authors provide a rigorous theoretical framework, introduce novel metrics, and conduct evaluations to demonstrate the limitations of current LLMs in abstract reasoning. The findings somewhat highlight key areas for future research.

**Essential References Not Discussed:**

The paper provides a good review of related work in abstract reasoning and evaluation of large language models (LLMs). However, there are a few works that could further enrich the context and provide additional depth to the discussion:

1. The integration of neural networks with symbolic reasoning is an active area of research. The paper could reference Neuro-Symbolic AI: The Third Wave by [Artur d'Avila Garcez, 2020], which discusses the combination of neural and symbolic approaches to improve reasoning capabilities.

2. The paper's focus on benchmarking abstract reasoning could be complemented by citing classic cognitive science benchmarks. For example, the Raven's Progressive Matrices have long been used to evaluate human abstract reasoning. Including such benchmarks would provide a broader perspective on the challenges and methods used in evaluating reasoning. (Using the text-based RPM problem in [Taylor Webb, Nature Human Behaviour, 2023])

3. This paper could cite "The Measure of Intelligence by [Chollet'19]", which provides a comprehensive framework for evaluating intelligence in AI systems. This work offers valuable context for the theoretical underpinnings of the paper's approach to abstract reasoning.

**Experimental Designs Or Analyses:**

The experimental designs and analyses in the paper are sound and valid. The authors evaluated various LLMs using a comprehensive benchmark with both Direct Prompting and Chain-of-Thought strategies. The results reveal some limitations in abstract reasoning and memory dependence. The findings may be helpful for improvement in LLMs.

**Methods And Evaluation Criteria:**

The methods and evaluation criteria in the paper are well-suited for assessing abstract reasoning in LLMs. The theoretical framework and metrics measure abstraction versus memorization. The benchmark, with systematic symbol remapping and diverse tasks, rigorously evaluates reasoning abilities. Overall, the proposed methods offer valuable insights for advancing abstract reasoning in LLMs.

**Other Comments Or Suggestions:**

N/A

**Other Strengths And Weaknesses:**

+:

1. The benchmark is designed to cover a wide range of tasks. This allows for a good view of different facets of abstract reasoning.

2. This paper introduces both "Abstract Reasoning Score" and "Memory Dependence Score" to provide a nuanced evaluation of model performance

3. The paper provides a strong theoretical foundation for abstract reasoning by formalizing the processes of abstraction and reasoning.

4. The paper includes extensive evaluations across various LLMs (7B-70B scale, API-based models, and agent frameworks).

-:

1. The paper acknowledges the influence of training data on model performance but does not provide a detailed analysis of how different training data configurations might affect abstract reasoning capabilities.

2. The paper does not provide a direct comparison of LLM performance to human-level reasoning.

3. The paper could provide more detailed implementation details for the benchmark tasks, such as specific examples of symbol remapping and task configurations.

**Questions For Authors:**

See Strengths And Weaknesses

**Relation To Broader Scientific Literature:**

The paper integrates cognitive science theories of abstraction and reasoning into a novel framework, extending prior work on generalization and reasoning evaluation. Its metrics and benchmark design address some limitations of existing methods, focusing on symbolic tasks and diverse reasoning facets. Empirical findings align with known challenges in LLMs' generalization and highlight the need for improved reasoning capabilities, advancing the discourse on AI abstract reasoning.

**Theoretical Claims:**

The paper presents several theoretical claims supported by formal proofs, including Theorem 3.7 (validity of Γ for Rule-Given potential), Theorem 3.8 (validity of ∆ for Rule-Inductive abstraction), and Theorem 3.9 (score range interpretation). These proofs aim to establish the mathematical soundness of the proposed metrics. The proofs appear to be correctly formulated and logically consistent, providing a rigorous foundation for the metrics. The Law of Large Numbers is appropriately invoked in Theorem 3.7, and the invariance properties are well-argued in Theorem 3.8. The score interpretation in Theorem 3.9 is also logically structured.

---

> ### Author Rebuttal · Authors · 2025-04-01
>
> ## 1. Relevant References
>
> Thank you for highlighting these references. Chollet (2019) is already cited (see Introduction). In the revised version, we have added:
>
> - **Garcez (2020), *Neurosymbolic AI: The 3rd Wave***
>   This work highlights the limits of connectionist methods and supports our discussion on transitioning from memorization to true abstraction.
>
> - **Webb et al. (2023), *Emergent Analogical Reasoning in Large Language Models***
>   This study provides empirical evidence on zero-shot analogical reasoning and contextualizes our findings on performance drops under symbol remapping.
>
> ---
>
> ## 2. Analysis of Training Data Configurations
>
> ### Experimental Design Overview
>
> To further evaluate the influence of different training data configurations on abstract reasoning, we tested two models—**Llama-3.1-8B-Instruct (fine-tuned)** and **GEMINI-2.0-FLASH-THINKING-EXP-01-21**—and also tested human participants.
>
> **Training Data Configuration:**
>
> - **Unmapped (*) Configuration:**
>   The training data is generated using a random seed to follow the same distribution as the `fixed_len_chat_bit_raw_dataset`. No symbol mapping is applied.
> - **Fully Mapped (†) Configuration:**
>   The training data is generated using a random seed to follow the same distribution as the `fixed_len_chat_bit_dataset`. Symbol mapping is applied.
>
> **Test Benchmarks:**
>
> - We test on two mapped datasets:
>   - `fixed_len_chat_bit_dataset`
>   - `fixed_len_chat_str_dataset`
>
> **Training Setup:**
> Training was performed using a single NVIDIA A800 80GB GPU, with 2,000 training samples over 8 epochs, a batch size of 8, cosine learning rate scheduling (with 3% warm-up), and using bfloat16 mixed precision.
>
> ### Experimental Results and Analysis
>
> Below are our experimental results:
>
> | Model                                   | Untrained | Epoch 8 |
> |-|-|-|
> |  fixed_len_chat_bit_dataset|||
> | Llama-3.1-8B-Instruct*|0.13|0.18|
> | Llama-3.1-8B-Instruct cot*|0.05 |0.07|
> | Llama-3.1-8B-Instruct†|0.13|0.60|
> | Llama-3.1-8B-Instruct cot†|0.05|0.11|
> | gemini-2.0-flash-thinking-exp-01-21|0.57||
> | gemini-2.0-flash-thinking-exp-01-21 cot|0.46||
> | Human|0.97||
> | fixed_len_chat_str_dataset|||
> | Llama-3.1-8B-Instruct|0.27|0.27|
> | Llama-3.1-8B-Instruct cot|0.03|0.07|
> | Llama-3.1-8B-Instruct| 0.27|0.25|
> | Llama-3.1-8B-Instruct cot| 0.03|0.02|
> | gemini-2.0-flash-thinking-exp-01-21| 0.19||
> |gemini-2.0-flash-thinking-exp-01-21 cot|0.17||
> | Human|0.47||
>
>
>
> **Key Observations and Conclusions:**
>
> - Under the **unmapped configuration**, the performance improvement is marginal.
> - For the **fully mapped configuration**, the Llama-3.1-8B-Instruct model shows significant improvement on the bit dataset (from 0.13 untrained to 0.60 by Epoch 8). In contrast, the **Llama-3.1-8B-Instruct cot†** shows improvements from **0.05** to **0.11**. Analysis of the Chain-of-Thought outputs indicates that the model still largely imitates the training examples rather than explicitly inferring and applying abstract rules.However, the improvement on the string dataset is limited, suggesting that the rules learned under remapping do not generalize well across different symbolic representations.
> - **Human performance** exceeds that of the models, further underscoring that current LLMs heavily rely on memorized associations rather than fully extracting and applying abstract rules.
>
> ---
>
> ## 3. Comparison with Human-Level Abstract Reasoning
>
> We conducted a supplementary experiment with four undergraduate computer science students who evaluated the same tasks:
>
> - **Bit Dataset:** Human participants achieved an accuracy of 0.97.
> - **String Dataset:** The overall human accuracy was 0.47.
>
> This direct comparison clearly demonstrates that even with extensive training, LLMs continue to underperform relative to human abstract reasoning capabilities, reinforcing the need for further improvements in model design and training.
>
> ---
>
> ## 4. Detailed Benchmark Implementation
>
> We agree with this recommendation and will revise Appendix A.7 in a future version to include detailed examples demonstrating the symbol remapping process and specific task configurations. Furthermore, we commit to releasing the complete source code, which will comprise:
>
> - A dynamic symbol mapping tool with customizable remapping rules.
> - Code templates for generating custom datasets across all six defined task categories.
> - An automated evaluation pipeline that computes both Abstract Reasoning Score (Γ) and Memory Dependence Score (∆), ensuring consistent answer matching.
>
> ---
>
> Once again, we thank the reviewers for their highly constructive feedback and for helping us identify areas for clarification and improvement. We believe that the revisions and additional experiments described above have significantly strengthened our work, and we appreciate your careful consideration of our manuscript.
>
> Sincerely,
> The Author Team

---

> > ### Comment · Reviewer_GRws · 2025-04-02
> >
> > I sincerely appreciate the authors' responses. Most of my concerns have been addressed. I also hope the authors can incorporate the related studies and the analysis into the paper and discuss them. Considering that this work has some innovation and provides appropriate analysis, I am happy to maintain my original score.

---

> > > ### Author Response · Authors · 2025-04-06
> > >
> > > Dear Reviewer,
> > >
> > > Thank you very much for your careful review and valuable comments. We are delighted that you appreciate the innovations and analytical insights presented in our work, as well as our rigorous theoretical framework, experimental methodology, and overall benchmark design.
> > >
> > > In our revised manuscript, we have incorporated additional references on neurosymbolic integration and classical cognitive benchmarks—as mentioned in our previous rebuttal—and introduced new citations such as ​*Neural-Symbolic Learning and Reasoning: A Survey and Interpretation*​. We have also added a new experimental section, as outlined in our rebuttal, to compare unmapped versus fully mapped training data configurations, including human performance comparisons. This new section includes both a table and a line chart to clearly illustrate our findings. We have released the complete source code on GitHub.
> > >
> > > Once again, we sincerely thank you for your constructive feedback and for recognizing the value of our benchmark. Your positive remarks have been a great encouragement to our team, and we are confident that the revisions and additional experiments have further reinforced our theoretical contributions and research findings, as well as provided valuable insights for advancing abstract reasoning capabilities in LLMs.
> > >
> > >
> > > Sincerely,
> > > The Author Team

---

### Official Review · Reviewer_gYuP · 2025-03-22

**Overall Recommendation:** 4

**Summary:**

The goal of the paper is to evaluate the abstract reasoning capabilities of LLMs. The paper points out flaws with two existing benchmarking paradigms: The symbolic reasoning benchmarks like GSM8K risk memorization since the models could be (inadvertently) trained on these benchmarks. The visual abstract benchmarks like ARC are not well suited for LLMs due to their visual nature. The paper first lays out what it means by reasoning: abstraction, where the model identifies patterns from various inputs, followed by reasoning, where the model applies consistent rules to these abstractions to arrive at some outputs. The paper builds a symbolic benchmark and shows gaps in reasoning abilities of LLMs.

**Claims And Evidence:**

Yes. The main claim of the paper is the design of a formal setup for measuring abstract reasoning abilities of LLMs. The paper does a fairly good job of building this setup. The paper also does quite well in grounding this setup in existing literature in cognitive sciences.

**Essential References Not Discussed:**

Not that I could tell.

**Experimental Designs Or Analyses:**

I looked at the dataset design in Appendix A6. The datasets are relevant. However, the paper should add details and examples from these datasets. For instance, just by reading the name `var_len_chat_bitop_raw_dataset`, it is difficult for a reader to infer exactly what this dataset is and what the precise prompts used here were.

**Methods And Evaluation Criteria:**

Yes. The benchmarks are math related but do provide coverage across a large number of mathematical operations (Appendix 6).

**Other Comments Or Suggestions:**

None

**Other Strengths And Weaknesses:**

## Strengths

1. The fact that the paper formally defines the basic building blocks, e.g., concepts, abstraction and reasoning is quite useful and could lay a strong foundation for follow ups in this area.
2. The paper is quite well-written. Formalism (e.g., Definitions in Section 3.1) is followed up with concrete examples.


## Weaknesses
1. Definition 3.5: The paper should explicitly mention the probability distribution the test set T is samples from. Is it sampled from some task based distribution in the real world (e.g., SAT problems)? Could it contains duplicates that are the same problem but phrased differently?
2. Line 233: The paper should spend some more time on the validity of these alterations. Should we expect the same performance after altering 0/1 to A/B? One could argue both ways. For instance, humans can perform arithmetic in base 10 much more easily than in base, say 89. Does this different make humans poor abstract reasoners?
3. Theorem 3 proof in Appendix 2 needs some work. The proof verifies the properties for some extreme cases and shows some monotonicity. How does adherence to these properties make the metric in Eq. 9 a “valid” metric. What properties do we want in a valid metric anyway?

**Questions For Authors:**

1. ARC doesn’t work for LLMs, but would it work with multi-modal LLMs? Why not test multi-modal LLMs given that they are becoming quite ubiquitous. Also, humans learn from both language and vision. So why not consider multimodal datasets?
2. Line 113: “However, these often lack direct application to LLM abstract reasoning evaluation”. How does ConceptARC lack application to LLM abstract reasoning?
3. The same concept could be described in many different words. For instance, “four-legged”, “having four legs”, etc. Does the paper enforce some kind of minimality constraints on the concepts, e.g., all equivalent expressions of a concept should be mapped to a single word pr phrase.
4. Line 146, second column: On “For instance, with the abstract concept, …”. The example mentions that “dog is likely to bark”. The word “likely” here implies some form of uncertainty in the behavior, e.g., the dog might bark in 80% of cases upon seeing another dog. How does the framework plan to quantify this uncertainty. Should this quantification be a part of the rules?
5. Theorem 3.7: What is the probability in Eq. 7 computed over?
6. Line 913: The strings here like “JJJQQQQQ” seem quite challenging for a LLM tokenizer, and will lead to a large number of tokens as compared to natural language. Given the tokenization issues, should we expect LLMs to be able to solve the resulting problems?

**Relation To Broader Scientific Literature:**

The paper does a good job of framing itself w.r.t. the existing literature on LLM benchmarking.

**Theoretical Claims:**

I went over the proofs in Appendix 2. While the proofs algebraically seem correct, I think they need some work in connecting the algebra to the application. The paper should specify what exactly the probability distributions in these proofs are, how we sample from them, and what the properties like validity in Eq. 9 are.

---

> ### Author Rebuttal · Authors · 2025-04-01
>
> ### 1. Test Set Sampling and Duplicates
>
> 1. **Test Set Distribution:**
>    Uniform distribution.
>
> 2. **Real-World Distribution and Correlation:**
>    No, it is not sampled from a real-world task-based distribution such as SAT problems. However, our benchmark shows strong correlation with other benchmarks. For example, if we only consider the models tested in our paper (excluding agents):
>    gemini-2.0-flash-thinking-exp-01-21* achieved rank 1 with the highest Γ score in our benchmark, and on LiveBench its average reasoning score reached 78.17, outperforming all other models.
>    qwq-32B-preview* achieved scores comparable to 72B and API models in our benchmark, with a LiveBench average reasoning score of 57.71.
>    Qwen2.5-72B-Instruct consistently outperforms competing models both in our benchmark and on the Korean SAT LLM Leaderboard baseline.
>
> 3. **Elimination of Duplicates:**
>    No, duplicate phrasings have been strictly eliminated.
>
> ---
>
> ### 2. Symbol Remapping and Its Rationale
>
> 1. **Performance Consistency After Remapping:**
>    Yes. A truly abstract model should maintain comparable performance post-remapping. For instance, GEMINI-2.0-FLASH-THINKING-EXP-01-21 drops by 0.40 on the `fixed_len_chat_bit_dataset` after remapping, indicating sensitivity to symbolic changes. In contrast, four computer science undergraduates scored nearly identically (1.00 and 0.97).
>
> 2. **Humans and Non-Decimal Arithmetic:**
>
>    No. Although non-decimal bases may pose challenges, humans can perform arithmetic in various numeral systems once abstract rules are applied. In our test of 16 samples from the `add_base3_raw_dataset`, undergraduates achieved 0.87 accuracy, while *GEMINI-2.0-FLASH-THINKING-EXP* reached 0.75.
>
> ---
>
> ### 3. Validity of the Metric F(Γ, ∆)
>
> 1. **Properties of a Valid Metric:**
>    A valid metric should be monotonic (increasing with higher Γ and decreasing with higher ∆, sensitive (proportional to changes), continuous, and normalized within [0, 1].
>
> 2. **Our Metric:**
>   Defined as
>   F(Γ, ∆) = w1Γ + w2(1 − ∆)
>   it satisfies these properties.
>
> 3. **Justification:**
>    Our metric inherits the desirable mathematical properties documented in the literature (e.g., Sokolova and Lapalme).
>
> ---
>
> ### 4. Multimodal Considerations
>
> 1. ARC work for multi-modal LLMs
> Yes. While ARC is designed for 2D visual inputs and can potentially benefit multimodal LLMs. For example, Align-DS-V—an experimental vision-language model derived from DeepSeek-R1-Distill-Llama-8B—achieved a score of **40.5** on the ARC-Challenge (5-shot), compared to **21.4** from DeepSeek-R1-Distill-Llama-8B.
>
> 2. Consider multimodal datasets
> Our work complements ARC by isolating the text modality to evaluate abstract reasoning in LLMs. While multimodal LLMs are emerging and humans naturally integrate visual and linguistic cues, most LLMs excel at language processing. Focusing exclusively on text allows us to establish a clear baseline for abstract reasoning without the added complexity of visual data.
>
> ---
>
> ### 5. Applicability of ConceptARC to LLMs
>
> LLMs are optimized for one-dimensional text, not two-dimensional spatial data. Our experiments show that even when Arabic numerals and simple graphics are array-formatted, LLMs struggle with 2D spatial properties, making ConceptARC less suitable for assessing textual reasoning.
>
> ---
>
> ### 6. Minimality in Concept Representations
>
> Our abstraction mapping process as an information compressor that inherently normalizes equivalent expressions without requiring an explicit minimality algorithm.
>
> ---
>
> ### 7. Addressing Uncertainty in Reasoning
>
> The phrase “likely to bark” in our cognitive example was intended solely as an illustrative simplification. Our framework currently outputs deterministic results post-abstraction; future work will integrate probabilistic models to handle inherent uncertainties.
>
> ---
>
> ### 8. Clarification on Eq. 7's Probability
>
> In Theorem 3.7, the probability is computed over the test set T, where each task is independently sampled from a uniform distribution over the task space.
>
> ---
>
> ### 9. Tokenization and Long Symbolic Strings
>
> Sequences like “JJJQQQQQ” may split into multiple tokens by the tokenizer; however, modern LLMs handle these effectively. In our benchmark, the majority of token counts for such symbolic strings are below 5k—well within the 8K-token (or larger) context window of current models.
>
> ---
>
> ### 10. Clarification of Dataset Names
>
> - var_len: Variable operand lengths.
> - chat: Conversational style prompts.
> - bitop: Bit manipulation operations.
> - raw: Original symbols without remapping.
>
> Detailed descriptions and examples will be added to Appendix A.6.
>
> ---
>
> We appreciate the reviewer’s feedback and hope that these clarifications, along with our proposed modifications, demonstrate the robustness of our theoretical framework and benchmarking strategy.
> Thank you very much for your consideration.
>
> Sincerely,
> The Author Team

---

### Decision · Program_Chairs · 2025-05-01

**Decision:**

Accept (poster)

**Comment:**

This paper proposes a formal framework for studying consistent abstract reasoning, and applies it to evaluate a variety of LLMs.
After discussion, the majority of the reviewers generally agree that the paper offers a useful contribution to the literature, though various additions such as human baseline evaluations would make it a stronger and more well-rounded paper. On the whole, this work seems to offer some useful analysis, and is reasonably clearly presented.